# ImgCoT: Compressing Long Chain of Thought into Compact Visual Tokens for Efficient Reasoning of Large Language Model

Xiaoshu Chen [1]   Sihang Zhou [1]   Ke Liang [1]   Taichun Zhou [1]   Yaohua Wang [1]   Yang Gao [2]   Xinwang Liu [1]

## Abstract

Compressing long chains of thought (CoT) into compact latent tokens is crucial for efficient reasoning with large language models (LLMs). Recent studies employ autoencoders to achieve this by reconstructing textual CoT from latent tokens, thus encoding CoT semantics. However, treating textual CoT as the reconstruction target forces latent tokens to preserve surface-level linguistic features (e.g., word choice and syntax), introducing a strong linguistic inductive bias that prioritizes linguistic form over reasoning structure and limits logical abstraction. Thus, we propose ImgCoT that replaces the reconstruction target from textual CoT to the visual CoT obtained by rendering CoT into images. This substitutes linguistic bias with spatial inductive bias, i.e., a tendency to model spatial layouts of the reasoning steps in visual CoT, enabling latent tokens to better capture global reasoning structure. Moreover, although visual latent tokens encode abstract reasoning structure, they may blur reasoning details. We thus propose a loose ImgCoT, a hybrid reasoning that augments visual latent tokens with a few key textual reasoning steps, selected based on low token log-likelihood. This design allows LLMs to retain both global reasoning structure and fine-grained reasoning details with fewer tokens than the complete CoT. Extensive experiments across multiple datasets and LLMs demonstrate the effectiveness of the two versions of ImgCoT.

## 1. Introduction

Chain-of-Thought (CoT) (Wei et al., 2022; Chen et al., 2025b) reasoning enhances the reasoning ability of large language models (LLMs) but incurs substantial reasoning overhead due to lengthy intermediate traces (Sui et al., 2025).

To address this issue, recent studies propose latent CoT reasoning (Deng et al., 2024; Hao et al., 2024; Shen et al., 2025; Tan et al., 2025; Su et al., 2025), which aims to compress long textual CoT into a small set of latent tokens. Among them, autoencoder-based compression (Su et al., 2025) has shown strong reasoning performance. Typically, as illustrated in the left part of Figure 1, it applies an autoencoder (Hinton & Salakhutdinov, 2006; Vincent et al., 2008; Van Den Oord et al., 2017) framework, where an encoder maps textual CoT into compact latent embeddings, and a decoder is trained to reconstruct the original CoT from these latent tokens. Through this reconstruction objective, latent tokens are expected to encode the semantic information of the CoT, enabling LLMs to reason over compressed representations instead of explicit text.

Despite their success, existing approaches suffer from a fundamental limitation: they treat textual CoT as the reconstruction target. As a result, latent tokens are forced to preserve surface-level linguistic patterns, such as word choice, syntactic structures, and stylistic variations. This introduces a strong linguistic inductive bias, causing latent representations to prioritize language form rather than reasoning structure. The reconstructed outputs clearly reveal this core issue. As illustrated in the left part of Figure 1, although the trained autoencoder can reconstruct a linguistically fluent CoT from textual latent tokens, the recovered CoT often contains structural errors compared with the original CoT. For example, four rules in the original CoT are decoded into only three, with rule 4 omitted. Moreover, the reasoning dependency in rule 2 and rule 3 is incorrectly rewritten, altering the original logical flow. In general, latent tokens with linguistic inductive bias tend to memorize how reasoning is expressed instead of capturing how reasoning is organized, which hinders logical abstraction.

We argue that an effective latent reasoning representation should focus on reasoning structure rather than linguistic realization. To this end, we propose ImgCoT, which replaces textual CoT reconstruction with visual CoT reconstruction. Specifically, as shown in the middle part of Figure 1, we render textual CoT into images that explicitly encode the

[1] the College of Computer Science and Technology, National University of Defense Technology [2] Nanjing University. Correspondence to: Xinwang Liu <xinwangliu@nudt.edu.cn>.

*Proceedings of the 43rd International Conference on Machine Learning*, Seoul, South Korea. PMLR 306, 2026. Copyright 2026 by the author(s).

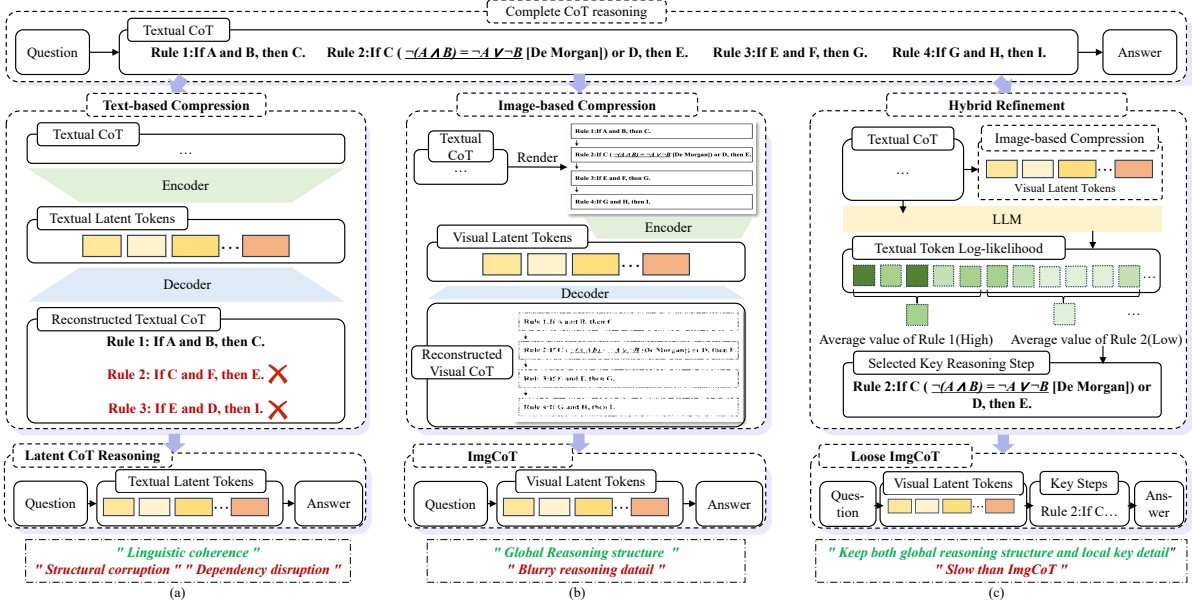

*Figure 1.* Comparison between three different latent reasoning methods. (a) The reconstructed textual CoT is linguistically fluent but contains structural errors compared with the original CoT, revealing that latent tokens in text-based compression tend to memorize expression patterns rather than reasoning organization. (b) The well-structured CoT in reconstructed visual CoT reflects that latent tokens in image-based compression preserve the global reasoning structure but blur the local reasoning details. (c) Loose ImgCoT further allocates a small number of additional tokens to allow LLMs to explicitly generate critical reasoning steps under uncertainty, thereby combining global reasoning structure with local reasoning precision.

layout of reasoning steps (e.g., step segmentation, hierarchical structure, and logical flow). Instead of reconstructing text, the autoencoder reconstructs these visual CoT. This design fundamentally changes the inductive bias of the compression process: text reconstruction enforces a linguistic inductive bias, while image reconstruction introduces a spatial inductive bias, encouraging the model to capture global reasoning layouts rather than word-level details. Thus, ImgCoT enables latent tokens to better abstract the high-level reasoning flow, such as step dependencies and logical progression, while discarding unnecessary linguistic variations.

However, from the reconstructed visual CoT in the image-based compression shown in Figure 1, we observe that while visual latent tokens effectively encode global reasoning structures, they may obscure fine-grained reasoning details. For general-purpose knowledge, this blurring has a limited impact, as LLMs already possess strong prior mastery of such skills. However, it can substantially affect domain-specific reasoning skills, where precise operational details are crucial. For example, De Morgan's law in Rule 2 of Figure 1, a fundamental logical rule for simplifying expressions, is difficult to completely encode into visual latent tokens. This limitation may lead to degraded reasoning performance in logic-related domains.

To mitigate this issue, we further propose Loose ImgCoT (L-ImgCoT), a hybrid reasoning paradigm that augments visual latent tokens with a small number of textual reasoning steps. These steps are automatically selected based on low token log-likelihood, which identifies uncertain or difficult reasoning steps that the LLMs struggle to infer implicitly. This hybrid design strikes a balance between global abstraction from visual latent tokens and local precision from selected textual steps, allowing LLMs to retain essential reasoning details while still benefiting from substantial token reduction compared to full CoT generation.

Our contributions can be summarized as follows:

- We reveal that visual inductive bias is more suitable than linguistic inductive bias for compressing CoT, as it better preserves global reasoning structure.

- We propose ImgCoT, the first visual CoT compression framework, and L-ImgCoT, an extension that retains fine-grained reasoning details.

- Extensive experiments across different datasets and LLM backbones demonstrate the effectiveness of Img-CoT and L-ImgCoT.

## 2. Related Work

### 2.1. Latent CoT Reasoning

Compared with other CoT compression methods (Chen et al., 2025a; Kang et al., 2025; Li et al., 2026; Ma et al.,

2025), latent-space CoT reasoning, which avoids generating long textual reasoning traces during inference, has recently attracted significant attention. ICoT(Deng et al., 2024), Coconut (Hao et al., 2024), and CCoT (Cheng & Van Durme, 2024) pioneered the concept of latent reasoning. Specifically, ICoT leverages curriculum learning to gradually internalize explicit textual CoTs into the forward propagation of LLMs, while Coconut and CCoT compress long textual CoTs into continuous latent vectors via curriculum learning. Subsequent works, such as CODI (Shen et al., 2025) and SynAdapt (Wang et al., 2025), further distill the reasoning capability of complete CoT-supervised models into latent-reasoning LLMs, improving latent reasoning performance. Beyond curriculum learning, CoLaR (Tan et al., 2025) compresses CoTs by pooling token representations across those of textual CoT, while LightThinker (Zhang et al., 2025) employs attention mechanisms for CoT compression.

However, although these methods accelerate CoT reasoning, their performance still falls short of full textual CoT reasoning. Recently, Su et al. (Su et al., 2025) broke this limitation by achieving latent reasoning without sacrificing performance. Specifically, they adopt Vector-Quantized Variational Autoencoders (VQ-VAEs) (Van Den Oord et al., 2017; Razavi et al., 2019) to compress the prefix of full textual CoTs, and train LLMs to reason jointly over compressed latent tokens and remaining textual steps. Nevertheless, as we analyze before, linguistic inductive bias limits the preservation of abstract reasoning structures in latent tokens, leading to suboptimal performance. Moreover, since most textual CoT steps still participate in inference, the acceleration gain remains limited.

### 2.2. Image Tokenization

Learning compact representations for images (i.e., image tokeniztion) has long been a central topic in generative modeling. Early studies attempt the autoencoder (AE) (Hinton & Salakhutdinov, 2006; Vincent et al., 2008) paradigm, which learns to map high-dimensional images into compact latent representations and reconstruct them back. Building upon this foundation, Variational Autoencoders (VAEs) (Kingma & Welling, 2013) introduce probabilistic latent variables, enabling continuous distribution modeling and facilitating downstream generative frameworks. To further discretize the latent space and ease sequence modeling, VQ-VAEs (Van Den Oord et al., 2017; Razavi et al., 2019) replace continuous latents with codebook-based discrete tokens, establishing a cornerstone for modern image tokenization.

Subsequent works extend these foundations along multiple dimensions, including adversarial (Esser et al., 2021; Chang et al., 2022) and perceptual (Zhang et al., 2018) training objectives for improving visual fidelity, transformer-based tokenizers (Yu et al., 2021; Cao et al., 2023) for enhanced mod-

eling capacity, and advanced quantization strategies such as multi-stage (Lee et al., 2022; Zheng et al., 2022) or lookup-free (Yu et al., 2023b; Mentzer et al., 2023) discretization. Despite these improvements, most existing methods inherit a common design choice: encoding images into a 2D grid of local tokens, which biases each token toward local spatial content.

Recently, 1D tokenization (Yu et al., 2024; Beyer et al., 2025) has been proposed to learn a latent sequence in which each token captures global spatial information, enabling significantly higher compression ratios by exploiting long-range redundancy. In our work, to preserve global reasoning structure within latent tokens, we adopt a 1D tokenization model, TiTok (Yu et al., 2024), as the backbone for compressing visual CoTs.

## 3. Method

ImgCoT consists of three stages: (1) Visual text tokenization, (2) LLM training with visual latent tokens, and (3) LLMs inference in latent space. Furthermore, we introduce Loose ImgCoT, a hybrid reasoning strategy that augments visual latent tokens with selectively preserved textual steps to retain fine-grained reasoning details.

### 3.1. ImgCoT

#### 3.1.1. VISUAL TEXT TOKENIZATION

Given a text sequence, we first render it into a visual text image $I \in \mathbb{R}^{H \times W \times 3}$. Specifically, as shown in Figure 2, we segment the input text based on predefined delimiters (e.g., "\n"), wrap each segment with a bounding box, and annotate inter-segment dependencies using directed arrows. This rendering converts linguistic input into a spatial representation, enabling subsequent modeling to exploit structural layouts rather than surface linguistic forms.

We adopt TiTok (Yu et al., 2024) as the autoencoder to compress $I$ into discrete latent tokens. First, the visual encoder $\mathcal{E}$ maps the $I$ into continuous feature embeddings:

$$h = \{h_i\}_{i=0}^n = \mathcal{E}(I), h_i \in \mathbb{R}^d. \tag{1}$$

To obtain discrete representations, we maintain a learnable codebook:

$$\mathcal{C} = \{e_j\}_{j=0}^k, e_j \in \mathbb{R}^d. \tag{2}$$

Then, each latent embedding $h_i$ is quantized by nearest-neighbor lookup in the $\mathcal{C}$, i.e.,

$$z_i = argmin_{e_j \in \mathcal{C}} ||h_i - e_j||_2. \tag{3}$$

This process converts continuous embeddings into discrete visual tokens, forcing the model to express $I$ using a finite set of reusable visual prototypes. Since the quantization operation is non-differentiable, we adopt the straight-through

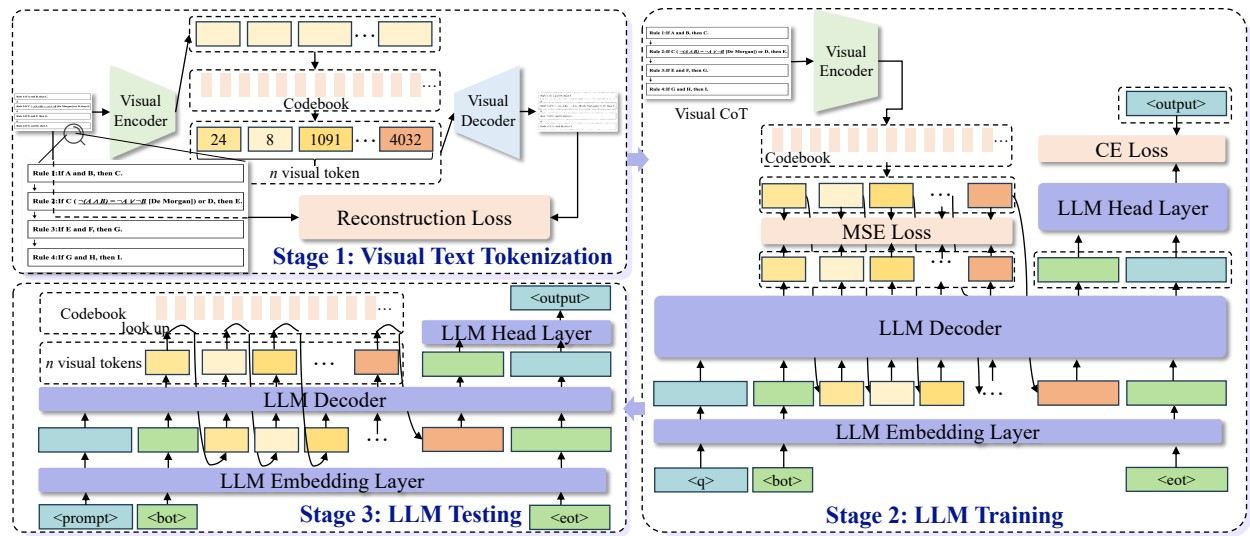

*Figure 2.* The overall training pipeline of ImgCoT. First, a visual encoder is trained by reconstructing visualized text, thereby compressing textual inputs into latent tokens. Then, the LLM is trained in an autoregressive manner to perform reasoning over the visual latent tokens. $\langle \cdot \rangle$ denotes the output sequence processed by the LLM tokenizer.

estimator to enable end-to-end training:

$$\hat{z}_i = h_i + sg(z_i - h_i), \quad (4)$$

where $sg$ denotes stop-gradient. The $\hat{z}_i$ are then fed into the visual decoder $\mathcal{D}$ to obtain reconstruct result $\hat{I} = \mathcal{D}(\hat{z})$. Finally, the reconstruction loss

$$\mathcal{L}_{rec} = ||\hat{I} - I||_2^2, \quad (5)$$

is applied to encourage the latent tokens $\hat{z}_i$ to capture the key information, such as the structural layout of the input text. Notably, this subsection only outlines the high-level framework, and further implementation details can be found in the TiTok (Yu et al., 2024).

### 3.1.2. LLM TRAINING WITH VISUAL LATENT TOKENS

The LLMs training process is shown in stage 2 of Figure 2. Specifically, let $q$ denote the question, $c$ the corresponding textual CoT, and $a$ the final answer. We first apply the visual encoder $\mathcal{E}$ and coodbook $\mathcal{C}$ learned in the first stage to compress $c$ into $z$, which consists of $n$ latent tokens. Meanwhile, we tokenize $q$ and $a$ using the LLM tokenizer, obtaining the token ids $\langle q \rangle = \{q_i\}_{i=0}^{|\langle q \rangle|}$ and $\langle a \rangle = \{a_i\}_{i=0}^{|\langle a \rangle|}$, respectively, where $\langle \cdot \rangle$ is the output sequence processed by the LLM tokenizer and $|\cdot|$ denotes the length of the input. Then, a training sample is constructed as

$$x = \{\langle q \rangle, \langle bot \rangle, z, \langle eot \rangle, \langle output \rangle\} = \{x_i\}_{i=0}^{|x|}, \quad (6)$$

where $\langle output \rangle = \langle a \rangle \oplus \langle eot \rangle$, $\langle bot \rangle \langle eot \rangle$ are the special token ids and $\oplus$ is the concatenation operation. Subsequently, we train the LLM in a standard autoregressive manner to

perform reasoning over latent tokens and generate the final answer, formulated as follows:

$$L_{ar} = \frac{\sum_{i=s_z}^{e_z} \mathcal{L}_{mse}(f_\theta(x_{:i}), x_i) + \sum_{i=s_o}^{e_o} \mathcal{L}_{ce}(f_\theta(x_{:i}), x_i)}{n + |\langle output \rangle|}, \quad (7)$$

where $s_z$ and $e_z$ denote the start and end indices of $z$ in the $x$, and $s_o$ and $e_o$ denote those of $\langle output \rangle$ in $x$. $f_\theta(x_{:i})$ represents the LLM output at the position $i-1$ given the input sequence $x_{:i}$, parameterized by $\theta$. Here, $x_{:i}$ denotes the subsequence of $x$ spanning indices from 0 to $i-1$. $\mathcal{L}_{mse}$ and $\mathcal{L}_{ce}$ denote the mean square error and cross-entropy loss, respectively.

Through this training procedure, the LLM learns to perform reasoning directly in the visual latent space. Since $n$ is significantly smaller than $|\langle c \rangle|$, the LLM can obtain the answer more efficiently than standard CoT reasoning.

### 3.1.3. TESTING PROCEDURE

As shown in stage 3 of Figure 2, inference is performed in an autoregressive manner. Importantly, the visual encoder is discarded at test time, ensuring that our method incurs no extra computational cost in real-world applications.

### 3.2. Loose ImgCoT

While visual latent tokens effectively capture global reasoning structure, they may obscure fine-grained domain-specific reasoning skills, thereby hindering LLMs from acquiring these skills during fine-tuning. To address this limitation, we propose L-ImgCoT, which augments visual latent tokens with a small set of critical textual reasoning steps.

L-ImgCoT follows the same training and inference procedures as ImgCoT. The only difference lies in the construction of training data. Following prior studies (Wan et al., 2024; Anonymous, 2023), the average log-likelihood of tokens in an input sequence reflects the LLM's confidence in understanding this input. Therefore, before fine-tuning, we compute the average token-level log-likelihood of the general pretraining corpus, denoted as $\gamma$, which represents the understanding level of LLMs to the general knowledge. Formally, the token-level log-likelihood at the position $i$ of an input text sequence $t$ can be evaluated as:

$$conf(t, i) = log(\text{softmax}(f_\theta(t_{:i}))^{\langle t_i \rangle}), \qquad (8)$$

where $\text{softmax}(f_\theta(t_{:i}))^{\langle t_i \rangle}$ denotes the probability predicted by LLMs that the token at position $i$ is $\langle t_i \rangle$. Based on it, we can get $\gamma$ by

$$\gamma = \sum_{t^j \in \mathcal{T}}^{|\mathcal{T}|} \sum_{i=0}^{|t^j|} conf(t^j, i), \qquad (9)$$

where $\mathcal{T} = \{t^j\}_{j=0}^{|\mathcal{T}|}$ is the general pretraining corpus.

Then, we use $\gamma$ as a threshold to filter each reasoning step $c^i$ in $c = \{c^i\}_{i=0}^{|c|}$. Specifically, a reasoning step $c^j$ is filtered out and be replaced with "..." if its average log-likelihood is greater than $\gamma$. Note that multiple consecutive filtered steps are replaced by a single ellipsis. After filtering, we obtain a refined reasoning trace $\hat{c} = \{\hat{c}^i\}_{i=0}^{|\hat{c}|}$. Here, $|\langle \hat{c} \rangle| < |\langle c \rangle|$.

As a result, we can replace the $\langle output \rangle$ in Equation (6) with $\langle output \rangle = \langle \hat{c} \rangle \oplus \langle a \rangle \oplus \langle eot \rangle$ to get the training data for L-ImgCoT. With this training data construction strategy, L-ImgCoT enables LLMs to reason primarily in the latent space for global structural abstraction, while selectively incorporating a small number of textual tokens to preserve domain-specific reasoning skills. This hybrid design achieves a favorable trade-off between reasoning efficiency and fine-grained logical precision.

# 4. Experiments

In this section, we first present the experimental setup in detail. We then systematically address the following key questions through the carefully designed experimental tasks:

- **RQ1:** How do our methods perform across different datasets and LLMs?

- **RQ2:** Why does image-based compression work better than text-based compression?

- **RQ3:** What other advantages does spatial inductive bias offer over linguistic inductive bias?

- **RQ4:** Does L-ImgCoT effectively preserve critical fine-grained reasoning skills during inference?

## 4.1. Experimental Setting

### 4.1.1. BENCHMARKS

We evaluate the proposed methods across three representative reasoning scenarios: mathematical reasoning, commonsense reasoning, and logical reasoning. For mathematical reasoning, we adopt GSM8K (Cobbe et al., 2021) and MATH (Hendrycks et al., 2021) as benchmarks. For commonsense reasoning, we evaluate our methods on GPQA-extended (Rein et al., 2024). For Logical reasoning, we use ProsQA (Hao et al., 2024), Last Letters Concatenation (LLC) (Kojima et al., 2022), and Date Understanding (DU) (Srivastava et al., 2023) to assess logical reasoning performance. More details of these dataset can be found in the Section A.1.

### 4.1.2. IMPLEMENTATION

Several key implementation details are below. More comprehensive descriptions are provided in the Section A.2.

1) Visual Text Tokenization. We train the autoencoder using MathPile (Wang et al., 2024) as the training corpus. The procedure for rendering text into images follows the implementation of Glyph (Cheng et al., 2025), as it provides the most suitable hyperparameters for rendering. The image resolution $H \times W$ is fixed to $512 \times 512$. Unless otherwise specified, we set the number of latent tokens to $n = 8$, the codebook size to $k = 4096$, and the latent dimension $d$ to be equal to the hidden size of the downstream LLM.

2) LLM Fine-tuning. Due to computational constraints, we adopt different fine-tuning strategies based on model size. For models with fewer than 1B parameters, we perform full-parameter fine-tuning. Otherwise, we apply LoRA (Hu et al., 2022) ($r = 12, \alpha = 32$) to conduct efficient fine-tuning.

## 4.2. RQ1 (Main Results)

### 4.2.1. COMPARISON WITH OTHER BASELINE METHODS

We compare ImgCoT and L-ImgCoT with the baseline Full-CoT and several recent, reproducible state-of-the-art (SOTA) latent reasoning methods, including Coconut (Hao et al., 2024), ICoT (Deng et al., 2024), CODI (Shen et al., 2025), and CoLaR (Tan et al., 2025), where Full-CoT supervises LLMs using complete textual CoT. The comparison results are summarized in Table 1. Note that, we implement Full-CoT ourselves, while all other SOTA methods are reproduced using the official implementation of CoLaR.

We observe that: (1) compared to other SOTA latent reasoning methods, ImgCoT achieves better and more stable performance under comparable reasoning costs. (2) Img-CoT matches or even outperforms Full-CoT in many cases, highlighting the importance of abstracting correct reasoning

*Table 1.* Comparison of accuracy (*Acc*) and token consumption of reasoning (# *Tokens*) across different methods. *w/* textual tokens denotes textual latent reasoning based on text compression, while *w/o* layout indicates directly rendering text into images without explicit logical dependency symbols (e.g., arrows). "-" represents that the result is missing due to the limitation of computational resources. Bold and underlined numbers indicate the best and second-best performance, respectively.

| Model | | MATH | | GSM | | GPQA | | ProsQA | |
|---|---|---|---|---|---|---|---|---|---|
| | | *Acc* ↑ | *# Tokens*↓ | *Acc*↑ | *# Tokens* ↓ | *Acc* ↑ | *# Tokens*↓ | *Acc*↑ | *# Tokens* ↓ |
| Qwen2.5-0.5B -Instruction | Full-CoT | 9.2 | 149.4 | 16.9 | 116.3 | 34.5 | 150.2 | 94.6 | 71.3 |
| | Coconut (COLM'25) | 3.5 | 6.0 | 3.7 | 6.0 | 26.0 | 6.0 | 73.8 | 6.0 |
| | ICoT | 3.3 | 0.0 | 3.9 | 0.0 | **39.6** | 0.0 | 54.9 | 0.0 |
| | CODI (EMNLP'25) | 0.4 | 6.0 | 0.5 | 6.0 | 27.1 | 6.0 | 0.0 | 6.0 |
| | CoLaR (NeurIPS'25) | 3.1 | 39.4 | 5.8 | 19.1 | 27.7 | 28.7 | 68.9 | 18.9 |
| | ImgCoT(ours) | 9.8 | 8.0 | 9.2 | 8.0 | 34.5 | 8.0 | 97.4 | 8.0 |
| | *w/o* layout | 9.1 | 8.0 | 9.2 | 8.0 | 32.7 | 8.0 | 96.8 | 8.0 |
| | *w/* textual tokens | 9.0 | 8.0 | 8.3 | 8.0 | 27.3 | 8.0 | 96.2 | 8.0 |
| | L-ImgCoT(ours) | **10.1** | 102.8 | **17.5** | 64.7 | 38.1 | 89.3 | **98.6** | 40.9 |
| Qwen2.5-1.5B -Instruction | Full-CoT | 19.4 | 238.7 | 44.1 | 126.6 | 40.0 | 229.5 | 99.6 | 46.2 |
| | Coconut (COLM'25) | 8.8 | 6.0 | 5.4 | 6.0 | 31.4 | 6.0 | 99.6 | 6.0 |
| | ICoT | 12.5 | 0.0 | 22.3 | 0.0 | 30.7 | 0.0 | 98.6 | 0.0 |
| | CODI (EMNLP'25) | 4.3 | 6.0 | 3.6 | 6.0 | 42.8 | 6.0 | 99.6 | 6.0 |
| | CoLaR (NeurIPS'25) | 4.9 | 60.5 | 6.9 | 22.9 | 37.1 | 30.8 | 99.4 | 7.1 |
| | ImgCoT(ours) | 19.5 | 8.0 | 38.7 | 8.0 | 41.8 | 8.0 | **100.0** | 8.0 |
| | *w/o* layout | 19.1 | 8.0 | 38.4 | 8.0 | 36.4 | 8.0 | **100.0** | 8.0 |
| | *w/* textual tokens | 17.4 | 8.0 | 37.5 | 8.0 | 36.4 | 8.0 | 99.2 | 8.0 |
| | L-ImgCoT(ours) | **19.9** | 127.4 | **45.2** | 71.3 | **43.6** | 130.8 | **100.0** | 21.7 |
| LLama3.2-3B -Instruction | Full-CoT | 23.8 | 205.9 | 60.5 | 105.6 | **45.5** | 179.3 | **100.0** | 41.3 |
| | Coconut (COLM'25) | 13.8 | 6.0 | 19.5 | 6.0 | 28.6 | 6.0 | **100.00** | 6.0 |
| | ICoT | 15.0 | 0.0 | 46.7 | 0.0 | 28.9 | 0.0 | 99.2 | 0.0 |
| | CODI (EMNLP'25) | - | - | 23.3 | 6.0 | - | - | - | - |
| | CoLaR (NeurIPS'25) | 19.1 | 41.2 | 27.4 | 20.1 | 20.4 | 33.9 | **100.0** | 6.8 |
| | ImgCoT(ours) | 24.1 | 8.0 | 56.8 | 8.0 | 43.6 | 8.0 | **100.0** | 8.0 |
| | *w/o* layout | 23.6 | 8.0 | 52.7 | 8.0 | 43.6 | 8.0 | **100.0** | 8.0 |
| | *w/* textual tokens | 22.6 | 8.0 | 49.3 | 8.0 | 41.8 | 8.0 | **100.0** | 8.0 |
| | L-ImgCoT(ours) | **24.3** | 143.8 | **61.0** | 63.2 | **45.5** | 115.7 | **100.0** | 27.5 |

structures during reasoning. In contrast, Full-CoT's token-by-token generation of complete reasoning traces is often vulnerable to minor local errors that propagate to incorrect final answers. (3) L-ImgCoT consistently outperforms Full-CoT across all settings while reducing inference cost by approximately 30%, mitigating ImgCoT's tendency to blur critical reasoning details and offering a practical alternative for scenarios where inference latency is less constrained.

### 4.2.2. ABLATION STUDIES

In Table Table 1, we present two key ablation studies. First, we compare visual latent reasoning (ImgCoT) with text-based latent reasoning (*w/* textual tokens) in terms of performance. The text-based latent reasoning model shares the same architecture and training strategy as ImgCoT, differing only in that its reconstruction target is textual CoT, which replaces the reconstruction loss in Equation (5) with the next-token prediction loss. The results show that visual latent reasoning consistently outperforms its textual counter-

part, suggesting that spatial inductive bias is more suitable than linguistic inductive bias for CoT compression. Considering that ProsQA may not be sufficiently challenging for evaluating reasoning improvements, as model performance is already close to 100%, we further evaluate the two methods on the LLC and DU datasets, as shown in Table 5 of the Appendix. The results further highlight the advantages of ImgCoT.

Second, we examine the impact of directly rendering textual reasoning steps into images (*w/o* layout) versus augmenting renderings with explicit symbolic cues (e.g., arrows) that encode reasoning logic (ImgCoT). We find that direct rendering yields slightly inferior reasoning performance compared to ImgCoT. This demonstrates the importance of accurately modeling logical dependencies among reasoning steps in the visual space, and further suggests that visual latent tokens indeed preserve structural dependencies between reasoning steps, as otherwise the quality of spatial dependency modeling for reasoning steps would not affect reasoning performance. More detailed ablation studies on explicit

symbolic representations can be found in the Appendix B.

## 4.3. RQ2 (Image-based Compression vs. Text-based Compression)

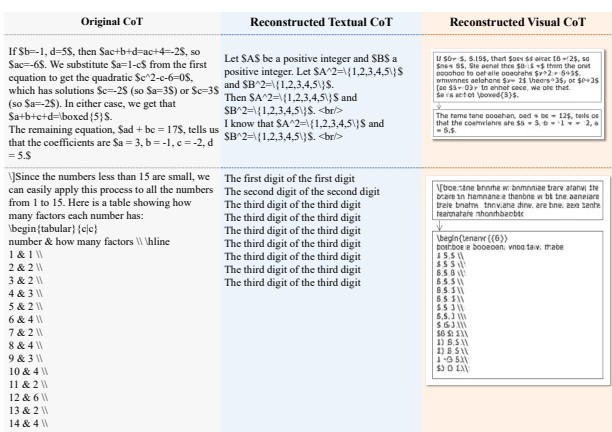

*Figure 3.* Qualitative comparison between the original CoT and its reconstructions from different latent representations. Text-based compression preserves surface-level linguistic forms, whereas image-based compression captures the global reasoning skeleton and stepwise layout of CoT. These examples illustrate the intrinsic inductive biases of different latent representations.

### 4.3.1. QUALITATIVE COMPARISON

We first provide a qualitative comparison. As shown in Figure 3, we visualize reconstructions decoded from latent tokens under text-based and image-based compression. We observe that (1) reconstructed textual CoTs remain linguistically fluent but exhibit weak semantic alignment with the original CoTs, indicating that text-based compression fails to preserve the underlying semantics of the original CoT. In contrast, reconstructed visual CoTs preserve both the logical dependencies across reasoning steps and the internal logic within individual steps, for example, faithfully recovering the direction of arrows in both cases and the exact table row count in the second reasoning step of the case shown in the second row. These reconstruction differences directly reflect the distinct types of information retained in latent tokens by text-based versus image-based compression: the former tends to encode surface-level linguistic form, whereas the latter prioritizes abstractions of reasoning structure. Since accurate modeling of reasoning structure is more critical for reasoning performance, image-based compression is therefore more effective for CoT compression.

### 4.3.2. QUANTITATIVE COMPARISON

We next present a quantitative comparison. As shown in Figure 4, we evaluate LLM reasoning performance as the number of latent tokens $n$ is gradually reduced under the two compression strategies. As $n$ decreases, latent tokens retain progressively less information, requiring each compression

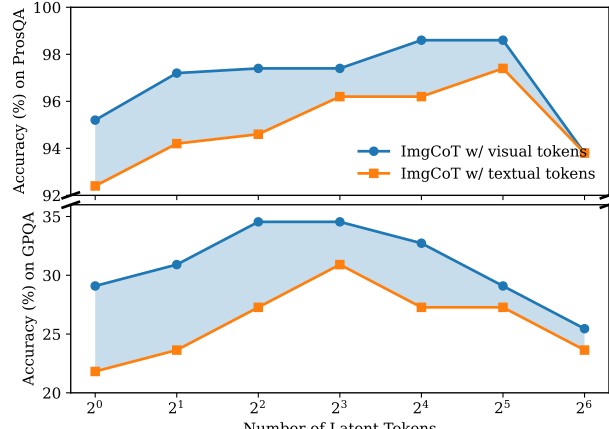

*Figure 4.* Performance comparison under varying numbers of visual versus textual latent tokens. The LLM is Qwen2.5-0.5B-Instruction.

method to prioritize which information is most critical to retain for effective reconstruction. We observe that as $n$ decreases, the performance gap between text-based and image-based compression steadily widens. This indicates that the information prioritized by image-based compression in latent tokens is more beneficial for LLM reasoning than that preserved by text-based compression, i.e, spatial inductive bias is better than linguistic inductive bias. We observe that model performance degrades when $n$ becomes very large. Due to space limitations, we defer a detailed analysis of this phenomenon to Section D.

## 4.4. RQ3 (Generalization of Spatial Inductive Bias)

In this subsection, we further demonstrate a key advantage of prioritizing structural dependencies in latent tokens (spatial inductive bias) over preserving surface-level lexical compositions (linguistic inductive bias): generalization. We argue that reasoning structures capture abstract logical patterns, which are more broadly transferable across reasoning tasks than task-specific lexical realizations. Accordingly, we compare the generalization ability of spatial and linguistic inductive biases, as shown in Table 2.

Specifically, we fine-tune LLMs on MetaMathQA (Yu et al., 2023a), while compressing its CoT using text-based or image-based compression, respectively. MetaMathQA is an augmented combination of the MATH and GSM datasets, and we therefore evaluate the resulting models on out-of-domain benchmarks, including Gaokao-Math-2023 (Liao et al., 2024), Svamp (Patel et al., 2021), MultiArith (Roy & Roth, 2015) and SingleEq(Koncel-Kedziorski et al., 2015). As shown in the table, visual latent reasoning with spatial inductive bias substantially outperforms text-based latent reasoning with linguistic inductive bias on all out-of-domain benchmarks, demonstrating the strong generalization capa-

*Table 2.* Out-of-domain generalization of visual and textual latent tokens at inference. The LLM is Qwen2.5-0.5B-Instruction.

| Model | In-Domain | | Out-of-Domain | | |
|---|---|---|---|---|---|
| | GSM | Gaokao | SVAMP | SingleEq | MultiArith |
| ImgCoT | 11.4 | 3.1 | 9.6 | 12.0 | 5.8 |
| *w/o* visual tokens | 10.0 | 1.8 | 5.6 | 8.4 | 4.2 |
| Difference | ↓ 1.4 | ↓1.3 | ↓3.0 | ↓3.6 | ↓ 1.6 |

*Table 3.* Performance Comparison of L-ImgCoT With and Without Visual Latent Tokens. The LLM is Qwen2.5-0.5B-Instruction.

| Method | MATH | GSM | GPQA | ProsQA |
|---|---|---|---|---|
| L-ImgCoT *w/o* visual tokens | 9.1 | 15.8 | 29.1 | 94.2 |
| L-ImgCoT *w/* visual tokens | 10.1 | 17.5 | 38.1 | 98.6 |

bility of spatial inductive bias in reasoning tasks.

### 4.5. RQ4 (Fine-grained Reasoning with L-ImgCoT)

In this section, we first demonstrate the effectiveness of the proposed strategy for retaining key reasoning steps. As illustrated in Figure 5 (additional examples are provided in the Section C), we observe that the retained reasoning steps correspond to essential domain-specific reasoning skills, such as writing code to generate plots (blue example) and applying mathematical theories (yellow example). These skills are critical for general-purpose LLMs to acquire, thereby validating the effectiveness of the proposed key-step retention strategy. In addition, we analyze the proportion of retained reasoning steps across different difficulty levels on the MATH dataset. More difficult problems typically involve a larger number of critical reasoning steps, and the retention ratios shown in the figure reflect this trend, supporting the rationality of the proposed retention strategy. Beyond training examples, the figure also presents the actual reasoning behavior of LLMs after L-ImgCoT training. We observe that the fine-tuned LLMs learn to retain only key reasoning steps, for example, outputting only empirical laws (red text) in mathematics.

Second, we demonstrate the importance of inserting visual latent tokens before key reasoning steps. As shown in the table, we compare LLM performance with and without visual latent tokens. We observe that LLMs perform substantially worse without visual latent tokens than when they are included. We attribute this difference to the fact that visual latent tokens encode global reasoning structure, which plays a critical role in guiding subsequent reasoning. Analogously, when humans form a global reasoning plan in advance, many

low-level details can be omitted, and only the outcomes of key reasoning steps need to be made explicit. In contrast, without such global abstraction, each individual reasoning step becomes necessary and cannot be safely omitted.

### 5. Limitations

Despite the strong performance and compression efficiency of ImgCoT, several limitations remain. Although our method preserves reasoning structure more effectively than text-based latent compression, the compressed visual latent representations are still not fully interpretable, as current image-CoT reconstructions may lose fine-grained symbolic and logical details. This limitation may become more pronounced in domains requiring precise symbolic reasoning. In addition, ImgCoT introduces extra rendering and visual encoding complexity compared to standard text-only reasoning pipelines. Nevertheless, unlike fully opaque latent reasoning approaches, ImgCoT retains the possibility of reconstructing compressed reasoning traces into human-understandable visual CoT, providing a practical compromise between reasoning efficiency and interpretability. Future work will focus on improving reconstruction fidelity and developing more structure-aware latent reasoning representations.

### 6. Conclusion

In this work, we revisit CoT compression through the lens of inductive bias and show that what is preserved in latent representations matters more than how much information is retained. Our key finding is that visual inductive bias, which prioritizes global reasoning structure over surface-level linguistic form, is fundamentally more suitable than linguistic bias for compressing CoT. Based on it, we propose ImgCoT, the framework that compresses CoT into visual latent tokens, and L-ImgCoT, an extension that further retains fine-grained reasoning details under lower inference budgets. Extensive experiments across multiple datasets and LLM backbones demonstrate the effectiveness of our proposed methods.

Looking forward, our work opens several directions, including more expressive structure-aware rendering schemes, richer visual elements beyond textual CoT, and more stable

**ImgCoT: Compressing Long Chain of Thought into Compact Visual Tokens for Efficient Reasoning of Large Language Model**

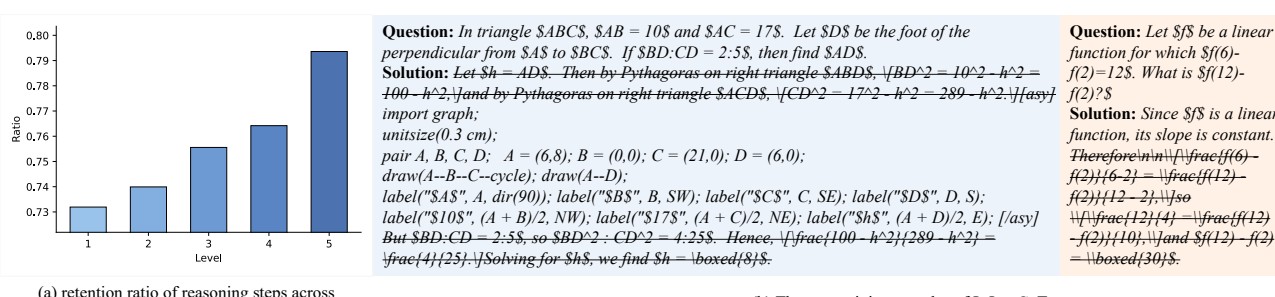

(a) retention ratio of reasoning steps across varying problem difficulty levels

(b) The two training samples of L-ImgCoT

**Question:** *What is the sum of the distinct prime factors of 735?*
**Solution by L-ImgCoT:** *[1] [2] [3] [4] [5] [6] [7] [8] ...~~A number is divisible by 3 if the sum of its digits is divisible by 3~~. ...Now, we need to factorize 245...Therefore, the complete prime factorization of 735 is:\n\n\\[ 735 = 3 \\times 5 \\times 7 \\times 7 \\]...#### 15*

(c) Reasoning result of LLMs training by L-ImgCoT

*Figure 5.* Effectiveness of Retaining Critical Reasoning Steps. The strikethrough text in (b) corresponds to reasoning deemed unimportant and filtered out by our strategy. [·] represents the latent tokens. The LLM adopted here is Qwen2.5-0.5B-Instruction.

training paradigms. We hope this work encourages future research to explore visual abstraction as a principled path toward efficient and generalizable reasoning.

## Acknowledgment

This work was supported by the National Science Fund for Distinguished Young Scholars of China (No. 62325604), the National Natural Science Foundation of China (No.62441618, No.62506371, and No.62276271), and the Major Program Project of Xiangjiang Laboratory (No. 24XJJCYJ01002).

## Impact Statement

This paper presents work whose goal is to advance the field of Machine Learning by studying more efficient and structured representations for CoT reasoning in LLMs. By examining the role of inductive bias in reasoning compression, our work contributes to improved efficiency and generalization of reasoning systems, which may help reduce reasoning cost and resource consumption in practical deployments.

There are many potential societal consequences of advances in efficient reasoning models, both positive and negative, which are well established in the literature on LLMs. We do not believe that this work introduces new ethical considerations beyond those commonly associated with improving model efficiency and reasoning capability, and therefore do not highlight specific societal impacts here.

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

**Question:** *Two boards, one four inches wide and the other six inches wide, are nailed together to form an X. The angle at which they cross is 60 degrees. If this structure is painted and the boards are separated what is the area of the unpainted region on the four-inch board? (The holes caused by the nails are negligible.) Express your answer in simplest radical form.\n\n[asy]\ndraw(6dir(150)--15dir(-30),linewidth(1));\ndraw((6dir(150)+12/sqrt(3)\*dir(30))--(15dir(-30)+12/sqrt(3)\*dir(30)),linewidth(1));\n\ndraw(6dir(210)-- (0,0),linewidth(1));\ndraw((9dir(210)+8/sqrt(3)\*dir(-30))--8/sqrt(3)\*dir(-30),linewidth(1));\n\ndraw(12/sqrt(3)\*dir(30)-- (12/sqrt(3)+6)\*dir(30),linewidth(1));\ndraw(12/sqrt(3)\*dir(30)+8/sqrt(3)\*dir(-30)--(12/sqrt(3)+9)\*dir(30)+8/sqrt(3)\*dir(-30),linewidth(1));\n\ndraw(2dir(150)— 2dir(150)+6dir(60),dashed);\ndraw(2dir(210)--2dir(210)+4dir(-60),dashed);\n\ndot((2,0));\ndot((4,-1));\ndot((8,1));\ndot((6,2));\n\nlabel(\"$60^{\\circ}$\", (11,1), E); \nlabel(rotate(30)\*\"$4^{\\prime\\prime}$\", .5\*(2dir(210)+2dir(210)+4dir(-60))+(0,-.5),W);\nlabel(rotate(-30)\*\"$6^{\\prime\\prime}$\", .5\*(2dir(150)+2dir(150)+6dir(60))+(1,1),W);\n[/asy]*

**Solution:** *Note that the unpainted region forms a parallelogram with heights between bases of 4 inches and 6 inches and with one angle 60 degree, as shown.\n\n*

*[asy]*
*size(150); unitsize(7.5,7.5); import olympiad;\ndraw(6dir(150)--15dir(-30),dashed);\ndraw((6dir(150)+12/sqrt(3)\*dir(30))--(15dir(-30)+12/sqrt(3)\*dir(30)),dashed);\ndraw(6dir(210)-- (0,0),dashed);\ndraw((9dir(210)+8/sqrt(3)\*dir(-30))--8/sqrt(3)\*dir(-30),dashed);\ndraw(12/sqrt(3)\*dir(30)--(12/sqrt(3)+6)\*dir(30),dashed);\ndraw(12/sqrt(3)\*dir(30)+8/sqrt(3)\*dir(-30)-- (12/sqrt(3)+9)\*dir(30)+8/sqrt(3)\*dir(-30),dashed);\n\nlabel(\"$60^{\\circ}$\",+(11,1),E,fontsize(8pt));\nlabel(\"$60^{\\circ}$\",+(9,1),+W,fontsize(8pt));\n\ndraw((0,0)--6/sin(pi/3)\*dir(30)-- (6/sin(pi/3)\*dir(30)+4/sin(pi/3)\*dir(-30))--4/sin(pi/3)\*dir(-30)--cycle, linewidth(1));\ndraw(4/sin(pi/3)\*dir(-30) -- (4/sin(pi/3)\*dir(-30) + 6/sin(pi/3)\*dir(60)));\ndraw(rightanglemark(4/sin(pi/3)\*dir(- 30),4/sin(pi/3)\*dir(-30) + 6/sin(pi/3)\*dir(30) + 4/sin(pi/3)\*dir(-30) + 6/sin(pi/3)\*dir(60),6/sin(pi/3)\*dir(-30) + 4/sin(pi/3)\*dir(-30) + 6/sin(pi/3)\*dir(60))/2,NW,fontsize(8pt));*
*[/asy]*

*The right triangle formed by drawing the height shown is a 30-60-90 triangle, and hence the hypotenuse has length $\\frac{6}{\\sqrt{3}/2} = 4\\sqrt{3}$ inches. Now considering the hypotenuse as the base of the parallelogram, our new height is 4, and thus the area of this parallelogram is $4\\cdot 4\\sqrt{3} = \\boxed{16\\sqrt{3}}$.*

**Question:** *Find all solutions $x$ of the inequality $$\\frac{5}{24} + \\left|x-\\frac{11}{48}\\right| < \\frac{5}{16}.$$Express your answer in interval notation, simplifying all fractions in your answer.*

**Solution:** ~~*We can make our work easier by rewriting all fractions in the inequality so that they have a common denominator of $48$: $$\\frac{10}{48} + \\left|x-\\frac{11}{48}\\right| < \\frac{15}{48}$$Then we subtract $\\frac{10}{48}$ from both sides: $$\\left|x-\\frac{11}{48}\\right| < \\frac{5}{48}$$The expression on the left side is the positive difference between $x$ and*~~ *$\\frac{11}{48}$. So, the inequality says that $x$ is strictly between $\\frac{11}{48}-\\frac{5}{48}$ and $\\frac{11}{48}+\\frac{5}{48}$. Simplifying these expressions and writing our answer in interval notation, we have $x\\in\\boxed{\\left(\\frac{1}{8},\\frac{1}{3}\\right)}$.*

**Question:** Two distinct integers, $x$ and $y$, are randomly chosen from the set $\\{1,2,3,4,5,6,7,8,9,10\\}$.  What is the probability that $xy-x-y$ is even?
**Solution:** ~~*We note that $xy-x-y$ is very close to the expansion of $(x-1)(y-1)$.*~~ *(This is basically a use of Simon's Favorite Factoring Trick.)\n\nIf $xy-x-y$ is even, then $xy-x-y+1 = (x-1)(y-1)$ is odd. This only occurs when $x-1$ and $y-1$ are both odd, so $x$ and $y$ must be even. There are $\\binom{5}{2}$ distinct pairs of even integers, and $\\binom{10}{2}$ distinct pairs of integers, so the probability is $\\dfrac{\\binom{5}{2}}{\\binom{10}{2}} = \\boxed{\\frac{2}{9}}$.*

*Figure 6.* The three training samples constructed for L-ImgCoT. Strikethrough indicates a reasoning step that is filtered out. The LLM adopted here is Qwen2.5-0.5B-Instruction.

# A. Experimental Setting

## A.1. Benchmarks

We evaluate the proposed method across three representative reasoning scenarios. Mathematical reasoning is first considered and evaluated on GSM8K (Cobbe et al., 2021) and MATH (Hendrycks et al., 2021). GSM8K consists of 8.5K grade-school arithmetic problems, while MATH contains 12.5K challenging competition-level problems. We next evaluate commonsense reasoning on GPQA-Extended (Rein et al., 2024), a scientific question-answering benchmark comprising 546 questions across biology, physics, and chemistry. Finally, we assess logical reasoning using ProsQA (Hao et al., 2024), which contains 17,886 logical reasoning problems. We further evaluate our method on two more challenging logical reasoning tasks: Date Understanding (DU) (Srivastava et al., 2023) and Last Letter Concatenation (LLC) (Kojima et al., 2022). For GSM8K, MATH, and ProsQA, we follow the official training and test splits, while for GPQA-Extended, we adopt the same data split strategy as CoLaR (Tan et al., 2025). As for the split of DU and LLC, one can refer to Fine-tune-CoT (Ho et al., 2023).

## A.2. Implementation

All training and evaluation are conducted on 8 NVIDIA A100 GPUs. Below we provide additional training details beyond those described in the main text.

### A.2.1. VISUAL TEXT TOKENIZATION

First, we adopt the text-to-image rendering strategy from Glyph(Cheng et al., 2025). A key difference is that, to fully accommodate an entire CoT within a $512 \times 512$ image, we employ a dynamic font-size adjustment strategy. The default font size is set to 9; if the full CoT cannot be rendered at this size, we iteratively reduce the font size until the entire CoT fits within the image. When the font size reaches its minimum and the entire text still cannot be accommodated, we render the remaining content into additional images. Notably, this situation does not arise in the datasets considered in this paper; however, this mechanism can be applied in more extreme scenarios. Conversely, when substantial blank regions are detected, we increase the font size until the blank area occupies no more than 50% of the image.

Then, the pipeline for training VQVAE to conduct visual text tokenization largely follows the official implementation of TiTok (Yu et al., 2024), with both the encoder and decoder consisting of 24 layers transformers. The main differences are that the number of latent tokens $n$ is set to 8, and the embedding dimension $d$ of the codebook is aligned with that of the downstream LLM. Morevoer, due to computational constraints, we reduce the batch size in the official implementation to 24. Additional architectural and training details can be found in TiTok.

*Table 4.* The hyper-parameters setting of fine-tuning LLMs.

| Initial Learning Rate | Learning Rate Schedule | Optimizer | Batch Size | Training Epochs |
|:---:|:---:|:---:|:---:|:---:|
| 1e-5 | cosine_with_restarts
(warm up steps=all training step * 15%) | AdamW
($\beta_1 = 0.9, \beta_1 = 0.95,$
$weight\_decay = 0.1$) | 4 | 5
(50 for GPQA) |

### A.2.2. LLM FINE-TUNING

All relevant training hyperparameters are summarized in Table 4. In addition, Table 6 reports the values of $\gamma$ of different LLMs. To evaluate the effect of $\gamma$, we conduct two dedicated ablation studies below.

First, the $\gamma$ on multiple general-domain pretraining corpora is figured out in Table 7. The resulting values were highly consistent across different corpora for the same model, demonstrating that $\gamma$ is robust and largely corpus-agnostic. This stability enables consistent reasoning performance without dataset-dependent calibration.

Then, we vary $\gamma$ around its default value (Table 8), which is computed as the average token log-likelihood over a general pretraining corpus, and evaluated its impact on the retained-step ratio, preserved token count, and final reasoning performance. A higher $\gamma$ (i.e., looser filtering) retains more reasoning steps and increases token usage, while the final performance remains largely unchanged, indicating that additional low-confidence steps contribute little to reasoning quality. Conversely, a lower $\gamma$ (i.e., stricter filtering) removes more reasoning steps, leading to performance degradation once essential reasoning information is discarded. These results suggest that the default $\gamma$ naturally achieves a good balance between reasoning-step preservation and token efficiency, thereby justifying our corpus-based estimation strategy without requiring task-specific tuning.

**Question:** *What is $0^{(5^{(6431564)})}$?*
**Solution:** *<1> <2> <3> <4> <5> <6> <7> <8> $0^x=0$ for any value of x ...#### 0*

**Question:** *What is the value of the following expression: $100 - 99 +98 - 97 + 96 - 95 + \cdots + 4 - 3 +2 - 1$?*
**Solution:** *<1> <2> <3> <4> <5> <6> <7> <8> The given sequence can be rewritten as $(100-99) + (98-97) + (96-95) + \cdots + (4-3) + (2-1)$...#### 50*

**Question:** *What is the area of trapezoid $OBCD$ below?*
*[asy]\nsize(200);\ndefaultpen(linewidth(0.8));\nxaxis(\"$x$\",-4,10);\nyaxis(\"$y$\",-3,5);\ndot(Label(\"$O$\",align=SW),(0,0));\ndot(Label(\"$D(2,3)$\",align=NW),(2,3));\ndot(Label(\"$C(4,3)$\",align=NE),(4,3));\ndot(Label(\"$B(8,0)$\",align=S),(8,0));\ndraw((0,0)--(2,3)--(4,3)--(8,0));\n[/asy]*
**Solution:** *[1] [2] [3] [4] [5] [6] [7] [8] We can use the formula for the area of a trapezoid, which is given by $\frac{1}{2}(b\_1 + b\_2)h$... To find the height, we draw a line segment from point $A$ to the x-axis, creating a right triangle with legs of length $AD = 7$ and $CD = 2$. ...#### 15*

*Figure 7.* The two cases generated by LLM using L-ImgCoT. [·] represents the latent tokens. The LLM adopted here is Qwen2.5-0.5B-Instruction.

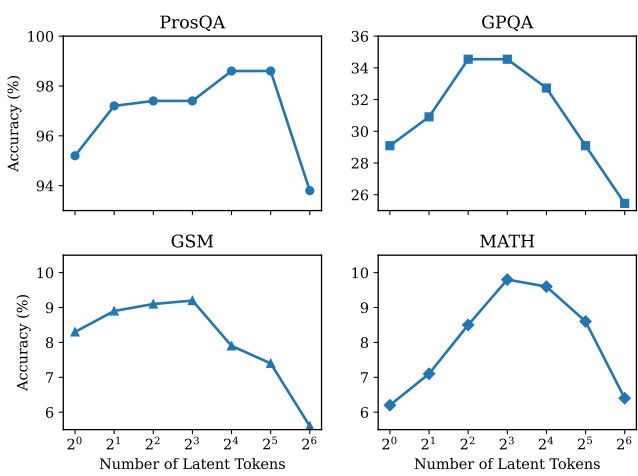

*Figure 8.* Trend of LLM Reasoning Performance with Varying Numbers of Latent Tokens. The LLM adopted here is Qwen2.5-0.5B-Instruction.

*Table 5.* Comparison of accuracy (*Acc*) and token consumption of reasoning (# *Tokens*) of different methods on more challenging logical reasoning tasks.

| Model | | LLC | | DU | |
|---|---|---|---|---|---|
| | | *Acc* ↑ | *# Tokens*↓ | *Acc*↑ | *# Tokens* ↓ |
| Qwen2.5-0.5B-Instruction | Full-CoT | 7.3 | 100.8 | 47.7 | 109.3 |
| | ImgCoT | **12.6** | 8.0 | **55.8** | 8.0 |
| | ImgCoT w/ textual tokens | 10.6 | 8.0 | 49.5 | 8.0 |
| Qwen-2.5-1.5B-Instruction | Full-CoT | **56.0** | 99.8 | 74.7 | 134.7 |
| | ImgCoT | 54.6 | 8.0 | **77.4** | 8.0 |
| | ImgCoT w/ textual tokens | 53.3 | 8.0 | 72.9 | 8.0 |
| Llama-3.2-3B-Instruction | Full-CoT | 76.6 | 104.9 | **81.1** | 106.6 |
| | ImgCoT | **78.6** | 8.0 | **81.1** | 8.0 |
| | ImgCoT w/ textual tokens | 77.3 | 8.0 | 78.3 | 8.0 |

*Table 6.* The value of $\gamma$ for different LLMs.

| Qwen2.5-0.5B-Instruction | Qwen2.5-1.5B-Instruction | LLama3.2-3B-Instruction |
|---|---|---|
| -1.58864506 | -1.35578818 | -1.76450461 |

*Table 7.* Default $\gamma$ across different corpora.

| Model | MathPile | Dolma (Soldaini et al., 2024) | Red-Pajama (Weber et al., 2024) |
|---|---|---|---|
| Qwen2.5-0.5B-Instruction | -1.58864506 | -1.64774219 | -1.72894371 |
| Llama3.2-3B-Instruction | -1.76450461 | -1.77974618 | -1.79100659 |

*Table 8.* Effects of different values of $\gamma$ on MATH under different LLMs.

| LLM | $\gamma$ | ACC | Retained-step (%) | #Token |
|---|---|---|---|---|
| Qwen2.5-0.5B-Instruction | -2.5886 | 9.8 | 47.6 | 77.6 |
| | -2.0886 | 9.7 | 59.4 | 89.4 |
| | -1.5886 (default) | 10.1 | 75.8 | 102.8 |
| | -1.0886 | 10.1 | 91.7 | 139.2 |
| | -0.5866 | 10.0 | 98.05 | 157.3 |
| Llama3.2-3B-Instruction | -2.76450461 | 23.8 | 48.9 | 117.9 |
| | -2.26450461 | 24.1 | 59.4 | 134.4 |
| | -1.76450461 (default) | 24.3 | 73.9 | 143.8 |
| | -1.26450461 | 24.3 | 89.9 | 186.2 |
| | -0.76450461 | 24.4 | 98.9 | 216.9 |

*Table 9.* Accuracy under different rendering settings of ImgCoT.

| Dataset | Model | ImgCoT | w/o segmentation | w/o boxes | w/ random arrows |
|---|---|---|---|---|---|
| MATH | Qwen2.5-0.5B-Instruction | 9.8 | 9.0 | 9.8 | 9.2 |
| GPQA | Qwen2.5-0.5B-Instruction | 34.5 | 30.9 | 34.5 | 32.7 |
| MATH | Llama3.2-3B-Instruction | 24.1 | 23.0 | 23.9 | 23.4 |

*Table 10.* Effect of XML Tag on text-based compression.

| Dataset | Model | w/ XML tag | w/o XML tag |
|---------|-------|------------|-------------|
| MATH | Qwen2.5-0.5B-Instruction | 9.0 | 9.0 |
| GPQA | Qwen2.5-0.5B-Instruction | 27.2 | 27.3 |
| MATH | Llama3.2-3B-Instruction | 22.8 | 22.6 |

## B. Mechanistic understanding of spatial inductive bias

We conduct targeted experiments comparing Image-CoT and Textual-CoT with and without explicit reasoning annotations.

For Image-CoT, reasoning steps are visually structured using segmentation, arrows, and bounding boxes. In Table 9, removing segmentation masks or randomizing arrows leads to noticeable performance degradation, while bounding boxes contribute only marginally.

We also annotate reasoning steps with XML tags and dependency markers in text-based compression. In contrast, in Table 10, removing these logical tags causes negligible performance changes, suggesting that latent tokens produced by text-based compression fail to effectively preserve reasoning structure.

These findings provide mechanistic evidence that ImgCoT preserves reasoning logic through spatial inductive biases, where segmentation and arrows explicitly encode hierarchical and inter-step relationships. In contrast, latent tokens generated by text-based compression appear insensitive to logical annotations, indicating that they do not faithfully capture the underlying reasoning structure.

## C. Case Study for the Ability to Preserve Reasoning Details

First, Figure 6 illustrates the validity of using $\gamma$ as a threshold to filter unimportant reasoning details. In the first case, the red-highlighted text shows that specialized reasoning skills, such as generating code for diagram construction, are preserved. The second case demonstrates that when no specialized reasoning skills are required, all reasoning steps can be filtered out under our strategy. The third case shows that our filtering strategy retains specialized skills such as mathematical formula transformations. Together, these cases collectively validate the effectiveness of the proposed filtering strategy.

Figure 7 further illustrates the behaviors exhibited by LLMs trained on the constructed dataset. In the first case, the LLM applies domain-specific knowledge in mathematical reasoning. The second case highlights the LLM's ability to retain critical details in algebraic formula transformations. In the third case, the LLM explicitly leverages key reasoning details, such as applying the trapezoidal formula and constructing auxiliary lines. Together, these cases demonstrate that LLMs trained with L-ImgCoT are able to preserve critical reasoning details during reasoning.

## D. Impact of Latent Token Count

We analyze how model performance varies with an increasing number of latent tokens, as shown in Figure 8. In principle, increasing the number of latent tokens allows more information from the original CoT to be preserved, which is expected to improve reasoning performance.

However, the empirical results reveal a non-monotonic trend, where performance first improves and then degrades as the number of latent tokens increases. We hypothesize that, when the number of latent tokens is small, increasing it indeed enriches the preserved CoT information and thus enhances performance. As the number of latent tokens further increases, the combinatorial space of latent representations grows rapidly; with a fixed training dataset, the model tends to underfit and fails to fully capture the underlying semantics. Consistently, when evaluating models with larger numbers of latent tokens, we observe that their latent-token-based reasoning often produces corrupted or incoherent outputs, which provides further evidence supporting our underfitting hypothesis. In future work, we plan to investigate how this trend evolves when the training data scale is increased.

