# OpenReview forum: "ImgCoT: Compressing Long Chain of Thought into Compact Visual Tokens for Efficient Reasoning of Large Language Model"
_ICML.cc/2026/Conference — ICML 2026 regular_

### Official Review · Reviewer_1JEu · 2026-03-08

**Soundness:** 2
**Presentation:** 2
**Significance:** 2
**Originality:** 2
**Overall Recommendation:** 4
**Confidence:** 2

**Summary:**

This paper introduces ImgCoT, a visual token-based approach to compressing long chains of thought for efficient reasoning in large language models (LLMs).

The authors argue that reconstructing visual renderings of CoTs, rather than textual CoTs, imparts a spatial inductive bias that better preserves the global reasoning structure as opposed to surface-level linguistic features.

The paper presents L-ImgCoT, a hybrid model that augments visual tokens with important textual steps identified by low log-likelihood, seeking to balance abstraction and detail.

Extensive experiments across multiple datasets and LLMs demonstrate that ImgCoT and L-ImgCoT achieve substantial reductions in reasoning token count, with competitive or superior reasoning accuracy compared to strong CoT-compression and full-CoT baselines.

**Compliance With Llm Reviewing Policy:**

Affirmed.

**Ethical Review Concerns:**

no concerns

**Key Questions For Authors:**

1. (Fairness of the experimental comparison) Did the model's improved performance stem from stronger "visual representation," or from the addition of "arrows" to the information provided? If similar logical markers are added to the text-based baseline (e.g., using XML tags `<step1>...</step1>` and annotating dependencies), will the visual token still have an advantage?

2. The paper mentions employing a strategy of dynamically reducing the font size to fit lengthy CoT statements into a 512 x 512 image.
When the inference steps are very long, the font size becomes extremely small. Despite the power of 1D Tokenizer (TiTok), can extremely small text pixel features still be preserved under high compression ratios?

3. How robust are the results to variations in the key threshold $\gamma$? It seems like the constant in your method. Do you observe performance sensitivity across datasets or models?

**Limitations:**

See questions.

**Strengths And Weaknesses:**

strengths:
- The paper clearly motivates that linguistic reconstruction in standard CoT compression can entangle reasoning content with language-specific realizations, hindering abstraction. The proposal to reconstruct visual representations instead is conceptually compelling.
- The technical pipeline is detailed, including the use of a strong 1D tokenization backbone (TiTok) and concrete log-likelihood-based filtering for hybrid retention of critical CoT steps.
- Results span four datasets (GSM8K, MATH, GPQA, ProsQA) and three LLM backbones, with substantial benchmarking against recent state-of-the-art CoT compressing models. Ablation studies are reported, and both qualitative and quantitative comparisons are provided.

weaknesses:

See questions.

---

> ### Author Rebuttal · Authors · 2026-03-29
>
> Thanks for your insightful comments. We address the key points below.
>
> ---
> ## 1.Fairness of Experimental Comparison
> We agree that ensuring a fair comparison between image-based and text-based representations is critical. To this end, we enhanced the text-based baseline by explicitly annotating reasoning steps with XML tags (e.g., \<step1\>…\</step1\>) and marking inter-step dependencies, mirroring the logical cues present in Image-CoT.
>
> Despite this enhancement, reasoning performance did not improve significantly compared to the standard text-based baseline and remained lower than Image-CoT (Table 1). This indicates that the advantage of image-based latent tokens arises not merely from explicit markers (arrows), but from the spatial inductive biases inherent in visual representations, which more effectively preserve hierarchical and relational reasoning structures.
>
> #### ## Table 1 Effect of XML tag.
>
> | Dataset | Model | w/ XML tag | w/o XML tag |
> |---|---|---|---|
> | MATH | Qwen2.5-0.5B-Instruction | 9.0 | 9.0 |
> | GPQA | Qwen2.5-0.5B-Instruction | 27.2 | 27.3 |
> | MATH | Llama3.2-3B-Instruction | 22.8 | 22.6 |
>
> ---
> ## 2. Preservation of Text Features under High Compression
>
> We appreciate the reviewer’s careful observation. As described in the Appendix, when the font size reaches a minimum or the reasoning trace exceeds the capacity of a single 512 × 512 image, the remaining content is rendered into additional images. Latent tokens from multiple images are concatenated to form a longer token sequence, ensuring all content is preserved. This mechanism prevents the font from becoming so small that it becomes incomprehensible to model.
>
> While this situation does not occur in any datasets used in this paper, the mechanism can handle more extreme cases, maintaining the integrity of visual information under high compression.
>
> ---
> ## 3. Sensitivity and Stability of Threshold γ
> We conducted a dedicated ablation study to evaluate the effect of γ.
>
> ### (a) Sensitivity to γ (Table 2 and 3):
> We varied γ around its default value, computed as the average token log-likelihood over a general pretraining corpus, and measured its effect on retained-step ratio, preserved token count, and final reasoning performance.
>
> - Higher γ (looser filtering) retains more steps, increasing token usage, but performance remains largely unchanged—additional low-confidence steps do not improve reasoning.
>
> - Lower γ (stricter filtering) reduces retained steps and performance drops if essential reasoning information is lost.
>
> **Thus, the default γ naturally balances sufficient reasoning-step retention and token efficiency, justifying its pretraining-corpus-based computation without per-task tuning.**
>
> ### (b) Stability across corpora (Table 4):
> We computed γ from multiple general-domain pretraining corpora. Across corpora, γ values for the same model were nearly identical, demonstrating robustness and corpus-agnostic stability. This ensures consistent reasoning performance without task-specific tuning.
>
> **Together, these results demonstrate that Image-CoT’s superior performance stems from the spatial inductive biases of visual latent representations rather than annotation style, that our compression preserves full reasoning content even under extreme conditions, and that γ provides a robust, generalizable threshold for reasoning-step selection.**
>
>
> #### Table 2 Effects of different values ​​of  γ. Dataset=MATH and LLM=Qwen2.5-0.5B-Instruction.
>
> | γ | ACC | Retained-step (%) | \#Token |
> |---|---|---|---|
> | -2.5886 | 9.8 | 47.6 | 77.6 |
> | -2.0886 | 9.7 | 59.4 | 89.4 |
> | **-1.5886 (default)** | **10.1** | **75.8** | **102.8** |
> | -1.0886 | 10.1 | 91.7 | 139.2 |
> | -0.5866 | 10.0 | 98.05 | 157.3 |
>
> ---
>
> #### Table 3 Effects of different values ​​of  γ. Dataset= MATH and and LLM=Llama3.2-3B-Instruction.
> | γ | ACC | Retained-step (%) | \#Token |
> |---|---|---|---|
> | -2.76450461 | 23.8 | 48.9 | 117.9 |
> | -2.26450461 | 24.1 | 59.4 | 134.4 |
> | **-1.76450461 (default)** | **24.3** | **73.9** | **143.8** |
> | -1.26450461 | 24.3 | 89.9 | 186.2 |
> | -0.76450461 | 24.4 | 98.9 | 216.9 |
>
> ####  Table 4 Default γ across different corpora
>
> | Model | MathPile | dolma | Red\-Pajama |
> |---|---|---|---|
> | Qwen2.5-0.5B-Instruction | -1.58864506 | -1.64774219 | -1.72894371 |
> | Llama3.2-3B-Instruction | -1.76450461 | -1.77974618 | -1.79100659 |
>
> ---
> **We hope this response can ultimately improve your overall assessment.**

---

### Official Review · Reviewer_jC9i · 2026-03-10

**Soundness:** 3
**Presentation:** 2
**Significance:** 3
**Originality:** 2
**Overall Recommendation:** 4
**Confidence:** 3

**Summary:**

This paper studies latent chain-of-thought (CoT) compression from the perspective of inductive bias. The paper argues that existing text-based latent compression methods tend to preserve surface linguistic form rather than reasoning organisation, whereas reconstructing a visualised CoT introduces a spatial inductive bias that better preserves global reasoning structure. Based on this idea, the paper proposes ImgCoT, which renders textual CoT into structured visual layouts and compresses them into a small number of visual latent tokens, and L-ImgCoT, a hybrid extension that preserves a small subset of textual reasoning steps selected by token log-likelihood. The main empirical claim is that image-based latent compression outperforms text-based latent compression at similar reasoning-token budgets, while L-ImgCoT further improves answer accuracy by adding back a limited amount of explicit reasoning detail.

**Compliance With Llm Reviewing Policy:**

Affirmed.

**Final Justification:**

I also do some prior work in latent cot, I think the key core is how to get a better representation. The author tell us some insight from the vision side, which is good and meaningful. The only thing I am concern is I think ProsQA is not sutiable for the evaluation since nearly all evaluation is near 100%, I hope in the next version the author could add some datasets. However, according to the rebuttal and other reviews, I believe the author could do this and I am looking forward to the next version. I will increase the score to 4.

**Key Questions For Authors:**

1. Clarification of L-ImgCoT’s inference behaviour. (weakness 1) My reading is that L-ImgCoT maybe does not perform online log-likelihood-based selection at inference time. Instead, key reasoning steps are selected during training-data construction, and the fine-tuned model is then trained to explicitly generate these steps at test time. If this interpretation is correct, I strongly suggest clarifying it more explicitly, since it is essential for understanding both Table 1 and the practical trade-off between ImgCoT and L-ImgCoT.

2. Reconciling “same inference procedure” with much higher token consumption.(weakness 2) The paper states that L-ImgCoT follows the same training and inference procedures as ImgCoT, but Table 1 shows a much larger reasoning-token budget for L-ImgCoT. Could the authors clarify this discrepancy more explicitly? My current understanding is that the input remains latent-token-based, but the decoded output now includes retained key reasoning steps before the answer. If so, this should be stated directly. In addition, the author might need to explain what the test token includes. Does it include only explicit text tokens or include them together with implicit tokens? Little confused why ICoT is 0.0.

3. Mechanistic understanding of spatial inductive bias. The paper provides evidence that image-based compression works better than text-based compression, but could the authors offer a deeper explanation of what exactly is preserved in visual latent tokens that is lost in text-based latent tokens? This would strengthen RQ2 beyond qualitative illustration and performance trends.

**Limitations:**

yes

**Strengths And Weaknesses:**

**Strength**

1. The paper identifies a clear and meaningful research gap, and the core idea is to some degree conceptually appealing.
The motivation is focused: prior latent CoT methods may compress reasoning effectively, but the reconstruction target itself may impose an inductive bias that affects which reasoning information is preserved. Framing the problem through linguistic vs. spatial inductive bias is a good conceptual contribution. In addition, the method ImgCoT/Loose CoT differs meaningfully from prior latent CoT work by reconstructing visualised CoT rather than textual CoT. This is a clear design that aims to preserve the global reasoning structure rather than rely on surface-level wording.

2. The main empirical results are reasonably comprehensive.
ImgCoT is generally competitive with or better than several latent reasoning baselines under similar reported reasoning-token budgets, and L-ImgCoT often improves further by recovering fine-grained details. In addition to the main results, the paper includes ablations on layout cues, a comparison between visual and textual latent compression, out-of-domain generalisation experiments, and analyses of retained reasoning steps in L-ImgCoT.

3. The visual-vs-textual comparison is supported well. The paper provides both qualitative evidence (e.g., reconstructed examples) and quantitative scaling trends with varying latent token counts, which together make the image-based compression claim more credible. I feel interested in section 4.5.

**Weakness**

1. The exposition of L-ImgCoT is not sufficiently clear, especially regarding its inference-time behaviour. The introduction may suggest that low log-likelihood triggers selective fallback to textual reasoning dynamically (figure 1 c), but the method section states that L-ImgCoT “follows the same training and inference procedures as ImgCoT” (line 220-221) and differs only in training-data construction. In practice, however, I think L-ImgCoT is trained to generate retained key reasoning steps explicitly at inference time? since in the Table 1, Loose CoT token is largely increased, which materially changes its behavior and explains its much higher token consumption in Table 1. This is an important point, but it is not explained clearly enough. Actually, I am confused for Loose CoT inference behavior at first time until I read Table 1.

2. It seems the efficiency comparison is not fully apples-to-apples. Table 1 reports “token consumption of reasoning,” but it appears to count explicit textual reasoning tokens rather than total computation, which might be why methods such as ICoT are shown as consuming 0.0 tokens while latent computation still clearly occurs. This makes the reported efficiency metric somewhat difficult to interpret, especially when comparing pure latent methods with hybrid methods such as L-ImgCoT. I am confused here and hope for an explanation.

3. Some benchmark choices are not fully aligned with the paper’s strong claim. The introduction emphasises logical abstraction as a central weakness of text-based compression, yet some evaluation settings, especially ProsQA for larger models, appear close to saturation, even 1.5b model with all methods nearly up to 100%, which limits their usefulness for diagnosing improvements in reasoning abstraction. In addition, I think the author needs to explain the rationality of their choice of this dataset, especially since they argue for logic ability or maybe consider other datasets.

4. The explanation of why image-based compression works better remains more empirical than mechanistic. The paper does provide evidence that image-based compression outperforms text-based compression in section 4.3, but the explanation still relies largely on intuition, examples, and scaling trends. The current RQ2 section shows that the approach works better, but does not yet fully explain (at least I am confused) why image-based compression could better preserve logical reasoning and what specific reasoning information is preserved by spatial inductive bias and lost under textual compression.

---

> ### Author Rebuttal · Authors · 2026-03-29
>
> We thank the reviewers for their insightful comments. Below we clarify the key points.
>
> ---
> ## 1.L-ImgCoT inference behavior
> **L-ImgCoT does not perform online log-likelihood–based selection at inference time. Instead, key reasoning steps are selected during training-data construction, and the model is trained to generate these retained steps—together with the visual latent tokens**. The whole process of L-ImgCoT  is:
>
> (a) Compute the average token log-likelihood γ  (Eq. 9).
>
> (b) Segment each CoT into step-level chunks; compute average log-likelihood per step.
>
> (c) Retain only lower log-likelihood steps than γ.
>
> (d) SFT LLMs to output visual latent tokens + retained reasoning steps + answer.
>
> (e) At inference, the model generates visual latent tokens + key steps + answer following the learned pattern.
>
> We will clarify this process in the revised manuscript.
>
> ---
> ## 2. Metric for Reasoning Efficiency
>
> (a) **“Reasoning token count" counts the generated tokens corresponding to the reasoning process (visual latent + textual reasoning tokens)**. This metric is widely adopted in recent latent-reasoning work (e.g., CoLaR, Coconut, Token Assorted) because, **under a standard autoregressive inference framework, the number of generated tokens is directly proportional to inference-time computational cost.**
>
> (b) **ICoT = 0.0 as it internalizes all reasoning traces, generating only answer.**  Because we only count the generated reasoning tokens (latent or textual), ICoT is shown as 0.0.
>
> (c) L-ImgCoT uses more tokens than ImgCoT because it outputs latent tokens + key reasoning steps + answer, whereas ImgCoT outputs only latent tokens + answer.
>
> (d) We agree with you that token count alone does not fully capture all aspects of inference efficiency. **To address this, we additionally measured inference time under identical settings to ensure fair efficiency comparison (Table 1)**
>
> #### Table1 Comparison of test time (ms) . LLM=Qwen2.5-0.5B-Instruction
> | Method | MATH | GSM | GPQA | ProsQA |
> |---|---|---|---|---|
> | Full-CoT | 616.6 | 589.7 | 619.2 | 565.9 |
> | Coconut | 116.7 | 82.3 | 75.6 | 110.6 |
> | ICoT | 61.77 | 32.0 | 10.9 | 51.06 |
> | CODI | 114.7 | 85.1 | 71.4 | 109.6 |
> | CoLaR | 400.5 | 207.2 | 313.8 | 233.5 |
> | ImgCoT (ours) | 150.2 | 98.8 | 89.6 | 134.8 |
> | L-ImgCoT (ours) | 582.1 | 552.4 | 577.2 | 402.7 |
>
> ---
> ## 3.Benchmark choice: ProsQA
> We selected ProsQA because it was released by the Meta team as a challenging logic-reasoning benchmark. This aligns with our goal of studying how different compression paradigms preserve logical reasoning structure.
>
> Even with high absolute accuracy at the 1.5B scale, text-based compression underperforms compared to image-based compression, revealing systematic differences in logical abstraction.
>
> **To better stress latent logical reasoning, we added experiments on StrategyQA, a dataset that evaluates a model’s ability to perform implicit multi-hop reasoning by requiring it to decompose a question into multiple latent facts and combine them logically. ** In Table 2, we observe a much larger performance gap, reinforcing that image-based compression better preserves logical structure.
>
> #### Table 2 Results on StrategyQA. LLM = Qwen2.5-0.5B-Instruction
>
> | Method | ACC |
> |---|---|
> | ImgCoT | 68.3 |
> | w/ textual tokens | 66.2 |
>
> ---
> ## 4.Mechanistic understanding of spatial inductive bias
>
> We conduct targeted experiments comparing Image-CoT and Textual-CoT with/without explicit reasoning annotations:
>
> ● Image-CoT (Table 3): reasoning steps are visually structured with segmentation, arrows, and bounding boxes. Removing segmentation or randomizing arrows degrades performance; bounding boxes contribute minimally.
>
> ● Textual-CoT (Table 4): reasoning steps are annotated with XML tags and dependency markers. Removing tags has a negligible effect, indicating text-based latent tokens largely fail to capture logical structure.
>
> **These results provide mechanistic evidence that ImgCoT preserves reasoning logic via spatial inductive biases—segmentation and arrows encode hierarchical and inter-step relationships. And latent tokens generated by text-based compression do not contain reasoning logical structure, as the performance is not sensitive to the logical tag.**
>
> #### Table 3 Accuracy under Different Rendering Settings. LLM=Qwen2.5-0.5B-Instruction
>
> | Dataset | Model | ImgCoT | w/o segmentation | w/o boxes | w/ random arrows |
> |---|---|---|---|---|---|
> | MATH | Qwen2.5-0.5B-Instruction | 9.8 | 9.0 | 9.8 | 9.2 |
> | GPQA | Qwen2.5-0.5B-Instruction | 34.5 | 30.9 | 34.5 | 32.7 |
> | MATH | Llama3.2-3B-Instruction | 24.1 | 23.0 | 23.9 | 23.4 |
>
> #### ## Table 4 Effect of XML tag.
>
> | Dataset | Model | w/ XML tag | w/o XML tag |
> |---|---|---|---|
> | MATH | Qwen2.5-0.5B-Instruction | 9.0 | 9.0 |
> | GPQA | Qwen2.5-0.5B-Instruction | 27.2 | 27.3 |
> | MATH | Llama3.2-3B-Instruction | 22.8 | 22.6 |
>
> ---
> **We hope this response can ultimately improve your overall assessment.**

---

> > ### Author Rebuttal · Reviewer_jC9i · 2026-04-03
> >
> > Thanks for author effect. I also do some prior work in latent cot, I think the key core is how to get a better representation. The author tell us some insight from the vision side, which is good and meaningful. The only thing I am concern is I think ProsQA is not sutiable for the evaluation since nearly all evaluation is near 100%, I hope in the next version the author could add some datasets. However, according to the rebuttal and other reviews, I believe the author could do this and I am looking forward to the next version. I will increase the score to 4.

---

> > > ### Author Response · Authors · 2026-04-03
> > >
> > > We deeply appreciate the reviewer’s positive feedback and confirmation that the raised issues have been addressed.
> > >
> > > We fully agree that ProsQA may not be sufficiently challenging for evaluating reasoning improvements. In response, we further evaluate our method on two more challenging logical reasoning tasks: Date Understanding (DU) and Last Letter Concatenation (LLC).
> > >
> > > Date Understanding (DU) requires the model to interpret natural-language date descriptions (e.g., “three days after the second Monday of July”) and compute the precise calendar date.
> > >
> > > Last Letter Concatenation (LLC) requires extracting the last letter of each word in a sequence and concatenating them into a new string.
> > >
> > > Both tasks involve multi-step symbolic logical reasoning and are known to be substantially more difficult than ProsQA. For details on data processing, please refer to [1].
> > >
> > > We include the full results in the table below (to be added in the revised manuscript). The results show:
> > >
> > > - DU and LLC remain far from being solved, even as model size increases—indicating they are meaningfully more challenging than ProsQA.
> > >
> > > - Our proposed image-based compression method (ImgCoT) consistently outperforms text-based compression (ImgCoT w/ textual tokens) on both tasks, demonstrating the robustness and effectiveness of our approach beyond ProsQA.
> > >
> > > We will incorporate these new experiments, detailed analysis, and the corresponding table in the revised version of the paper. We appreciate the reviewer’s encouragement and will continue strengthening the evaluation as suggested.
> > >
> > > ### Table Results on LLC and DU.
> > >
> > > | Model                           | Method                   | LLC-Accuracy | LLC-#Tokens | DU-Accuracy | DU-#Tokens |
> > > |---------------------------------|---------------------------|--------------|-------------|-------------|------------|
> > > | Qwen-2.5-0.5B-Instruction       | Full-CoT                  | 7.33         | 100.82      | 47.75       | 109.33     |
> > > | Qwen-2.5-0.5B-Instruction       | ImgCoT                    | 12.66        | 8.00           | 55.85       | 8.00          |
> > > | Qwen-2.5-0.5B-Instruction       | ImgCoT w/ textual tokens  | 10.66        | 8.00           | 49.54       | 8.00          |
> > > | Qwen-2.5-1.5B-Instruction       | Full-CoT                  | 56.00        | 99.85       | 74.77       | 134.74     |
> > > | Qwen-2.5-1.5B-Instruction       | ImgCoT                    | 54.66        | 8.00           | 77.48       | 8.00          |
> > > | Qwen-2.5-1.5B-Instruction       | ImgCoT w/ textual tokens  | 53.33        | 8.00           | 72.97       | 8.00          |
> > > | Llama-3.2-3B-Instruction        | Full-CoT                  | 76.67        | 104.93      | 81.08       | 106.68     |
> > > | Llama-3.2-3B-Instruction        | ImgCoT                    | 78.67        | 8.00           | 81.08       | 8.00         |
> > > | Llama-3.2-3B-Instruction        | ImgCoT w/ textual tokens  | 77.33        | 8.00           | 78.38       | 8.00          |
> > >
> > > [1] Ho et al. 'Large Language Models Are Reasoning Teachers,' in ACL 2023.

---

### Official Review · Reviewer_V6GB · 2026-03-12

**Soundness:** 3
**Presentation:** 3
**Significance:** 2
**Originality:** 2
**Overall Recommendation:** 4
**Confidence:** 3

**Summary:**

This paper studies latent chain-of-thought compression for LLM reasoning and argues that the standard design choice in prior work—compressing and reconstructing textual CoT—induces an overly strong linguistic inductive bias. The authors claim this causes latent tokens to preserve surface realization, such as wording and syntax, rather than the higher-level organization of reasoning. To address this, they propose ImgCoT, which renders CoT into a visual layout and trains a visual tokenizer/autoencoder to reconstruct the rendered CoT image instead of the original text. The core hypothesis is that a spatial inductive bias better preserves global reasoning structure, including step boundaries, dependency flow, and hierarchy. The method has three stages: first, textual CoT is rendered into structured images and compressed into discrete visual latent tokens using a TiTok-style tokenizer; second, the LLM is fine-tuned to answer questions conditioned on these visual latent tokens; third, inference is done directly in latent space, with the visual encoder discarded at test time. The paper further introduces L-ImgCoT (Loose ImgCoT), which supplements visual latent tokens with a small number of retained textual reasoning steps chosen via low token log-likelihood, aiming to recover fine-grained details that may be blurred by purely visual compression. Experimentally, the paper evaluates on four benchmarks spanning math, commonsense, and logic reasoning—GSM8K, MATH, GPQA-extended, and ProsQA—using three backbones: Qwen2.5-0.5B, Qwen2.5-1.5B, and Llama3.2-3B. The reported results suggest that ImgCoT is generally stronger than several latent reasoning baselines at similar token budgets, and that L-ImgCoT often matches or exceeds Full-CoT while reducing inference token usage relative to full textual reasoning. The paper also includes ablations against text-based latent compression and against an image rendering without explicit layout cues, plus an out-of-domain generalization study.

**Compliance With Llm Reviewing Policy:**

Affirmed.

**Final Justification:**

The rebuttal addresses my concerns well.

**Key Questions For Authors:**

NA

**Limitations:**

The authors should add a short limitations section that discusses some issues and limitations of their work. First, compressing reasoning into visual latent tokens may reduce interpretability and auditability, making it harder to inspect failures, verify safety-critical reasoning, or identify when the model has silently dropped crucial steps. Second, the method may unevenly degrade performance on specialized domains that require exact symbolic or procedural detail, which could be risky in education, science, medicine, law, or other high-stakes settings.

**Strengths And Weaknesses:**

Strengths:
1. ImgCoT consistently matches or outperforms Full-CoT baselines using only 8 latent tokens, a dramatic compression compared to typical reasoning trace lengths of 100–250 tokens. L-ImgCoT outperforms Full-CoT across nearly all settings while reducing token consumption by roughly 30–45%. The comparisons against strong baselines (Coconut, ICoT, CODI, CoLaR) are comprehensive, and the gap with these methods is substantial.
2. The paper carefully disentangles two distinct contributions: (a) image vs. text as the reconstruction target, and (b) the addition of layout structure (arrows and bounding boxes) vs. raw text rendering. Both ablations are internally consistent across three model scales, which increases confidence in the conclusions. The quantitative ablation in Figure 4 — showing widening performance gaps as token count decreases — is particularly compelling.
3. The out-of-domain generalization experiment (Table 2), where models fine-tuned on MetaMathQA are evaluated on Gaokao, SVAMP, MultiArith, and SingleEq, is a thoughtful addition. The consistent advantage of spatial inductive bias over linguistic bias on unseen task distributions supports the claim that spatial representations encode more abstract, transferable reasoning patterns.


Weaknesses:
1. The key-step selection in L-ImgCoT relies on a pre-computed γ threshold derived from a static general pretraining corpus. This is a domain-agnostic criterion that does not adapt to the specific question being asked at test time. Harder instances may require retaining different steps than easier ones, and a more dynamic or query-conditioned selection mechanism may be warranted. Figure 5(a) shows that retention ratios do vary by difficulty, but the selection itself is still governed by a global fixed threshold.
2. All three backbone models are relatively small (≤3B parameters). It remains unclear whether the observed benefits of spatial inductive bias hold for larger-scale models (e.g., 7B, 13B, 70B), where linguistic priors may be stronger and the dynamics of latent compression could differ substantially. The paper would be significantly stronger with at least one experiment at the 7B scale.

---

> ### Author Rebuttal · Authors · 2026-03-29
>
> We thank the reviewer for the constructive feedback and address the main concerns below.
>
> ## 1. On Static γ and Instance-Level Adaptation
> To examine whether a static γ limits adaptivity, we conducted three sets of analyses.
> ### (a) Comparison with instance-aware selection.
> We implemented an instance-aware selection strategy, skip-thinking [1], where step retention is dynamically determined for each sample. **Its performance is comparable to our γ-based filtering, indicating that the additional complexity of per-instance selection does not yield tangible improvements.**
> #### Table 1 Skip-thinking vs. L-ImgCoT (LLM = Qwen-2.5-0.5B-Instruction)
> | Method | MATH ACC | MATH #Token | GSM ACC | GSM #Token | GPQA ACC | GPQA #Token | ProsQA ACC | ProsQA #Token |
> |---|---|---|---|---|---|---|---|---|
> | skip-thinking | 9.4 | 116.4 | 17.4 | 87.6 | 38.1 | 109.2 | 97.8 | 50.1 |
> | L-ImgCoT | 10.1 | 102.8 | 17.5 | 64.7 | 38.1 | 89.3 | 98.6 | 40.9 |
>
> ### (b) Why a static γ still enables adaptive behavior.
> Although γ is fixed from a general corpus, the selection it induces is instance-sensitive as it depends on token log-likelihoods. As shown in Fig. 5(a) in the manuscript, harder instances contain more low–log-likelihood steps and therefore retain more steps. Thus γ is global, but the resulting selection is instance-dependent.
> ### (c) Sensitivity analysis (Table 2 and 3).
> We vary γ around its default value—the average token log-likelihood over general pretraining corpus—and measure (i) retained-step ratio, (ii) token count, and (iii) accuracy.
>
> ● Higher γ: More steps are retained. However, performance remains essentially unchanged,  indicating that including additional steps beyond a certain point does not further improve reasoning quality.
>
> ● Lower γ: Fewer steps are preserved. Once γ falls below the default value, accuracy drops, showing that informative steps are being removed.
>
> **These findings suggest that the default γ captures a stable boundary between sufficient and insufficient step retention. This supports using a corpus-based γ is already robust per task.**
> #### Table 2 Effects of different values ​​of  γ. Dataset=MATH and LLM=Qwen2.5-0.5B-Instruction.
> | γ | ACC | Retained-step (%) | \#Token |
> |---|---|---|---|
> | -2.5886 | 9.8 | 47.6 | 77.6 |
> | -2.0886 | 9.7 | 59.4 | 89.4 |
> | **-1.5886 (default)** | **10.1** | **75.8** | **102.8** |
> | -1.0886 | 10.1 | 91.7 | 139.2 |
> | -0.5866 | 10.0 | 98.05 | 157.3 |
> #### Table 3 Effects of different values ​​of  γ. Dataset= MATH and and LLM=Llama3.2-3B-Instruction.
> | γ | ACC | Retained-step (%) | \#Token |
> |---|---|---|---|
> | -2.76450461 | 23.8 | 48.9 | 117.9 |
> | -2.26450461 | 24.1 | 59.4 | 134.4 |
> | **-1.76450461 (default)** | **24.3** | **73.9** | **143.8** |
> | -1.26450461 | 24.3 | 89.9 | 186.2 |
> | -0.76450461 | 24.4 | 98.9 | 216.9 |
>
> ---
> ## 2.On Scaling to 7B LLM (Table 4)
> We evaluate ImgCoT on Qwen-2.5-7B. The results mirror the findings on ≤3B: the spatial inductive bias continues to provide consistent improvements. This confirms that ImgCoT generalizes beyond small backbones.
> #### Table 4 Full-CoT vs. ImgCoT vs. ImgCoT w/ Textual Tokens
> | Method | MATH ACC | MATH #Token | GSM ACC | GSM #Token |
> |---|---|---|---|---|
> | Full-CoT | 30.4 | 301.7 | 75.8 | 157.3 |
> | ImgCoT | 31.5 | 8 | 74.4 | 8 |
> | w/ textual tokens | 29.9 | 8 | 72.3 | 8 |
>
> ---
> ## 3. Limitations and Practical Considerations
> We will add a dedicated limitations section summarizing as follows:
> ### (a) Interpretability and auditability.
> Compressing reasoning into visual latent tokens may reduce direct interpretability. Although these tokens can be decoded back into image-based CoT, current reconstructions are not yet fully faithful. Improving reconstruction fidelity and readability is an important direction we will pursue.
> ### (b) Specialized domains requiring precise symbolic detail.
> In domains such as mathematics, science, medicine, and law—where precise symbolic reasoning is critical—our latent compression may unevenly degrade performance. Future improvements in high-detail image-CoT reconstruction may mitigate these issues by preserving finer symbolic cues.
> ### (c) Comparison with other latent-reasoning paradigms.
> **Unlike fully opaque latent reasoning methods, ImgCoT maintains the possibility of decoding latent tokens back into understandable visual CoT.** While reconstruction fidelity is imperfect, this provides a meaningful interpretability advantage compared to non-decodable latent methods. **We view L-ImgCoT as a practical step toward interpretable latent reasoning: it offers substantial compression while preserving a clear reasoning trace.** Until high-fidelity image-CoT decoding becomes available, this provides a interpretable compromise between efficiency and transparency.
>
> ---
> **We hope this response can ultimately improve your overall assessment.**
>
> [1] Chen et al. 'Skip-Thinking: Chunk-wise Chain-of-Thought Distillation Enable Smaller Language Models to Reason Better and Faster,' EMNLP

---

> > ### Author Rebuttal · Reviewer_V6GB · 2026-04-01
> >
> > Authors' reponses have addressed my concerns and I have raised my score accordingly.

---

> > > ### Author Response · Authors · 2026-04-01
> > >
> > > We sincerely thank the reviewer for the positive feedback and for acknowledging that the concerns have been addressed. The reviewer’s insightful comments have been very important for improving the quality of our paper. We are also greatly encouraged by the reviewer’s recognition and support, which motivate us to further strengthen our work in future research.

---

### Official Review · Reviewer_fFbf · 2026-03-13

**Soundness:** 3
**Presentation:** 3
**Significance:** 3
**Originality:** 3
**Overall Recommendation:** 4
**Confidence:** 4

**Summary:**

This paper studies how to compress long chains of thought into a compact latent representation for efficient reasoning in large language models. Instead of compressing textual CoT directly, the authors propose ImgCoT, which renders the reasoning process as an image and compresses it into a small set of visual latent tokens. Building on this idea, they further introduce L-ImgCoT, a hybrid variant that supplements the visual tokens with a few retained textual reasoning steps. The method is evaluated on multiple reasoning benchmarks and compared with prior latent reasoning baselines. The results show that the proposed approach can achieve better reasoning performance with substantially fewer inference tokens.

**Compliance With Llm Reviewing Policy:**

Affirmed.

**Key Questions For Authors:**

1. How sensitive is L-ImgCoT to the choice of the threshold γ? Please provide an ablation showing how changing γ affects the retention ratio of reasoning steps, the number of textual tokens preserved, and final reasoning performance.
2. What component of the layout contributes most to the observed gains? In the current ablation, “w/o layout” removes explicit symbolic cues such as arrows, but this does not isolate whether the benefit comes from arrows, segmentation into separate lines/boxes, or both. A more controlled ablation here would help validate the core claim that spatial inductive bias preserves reasoning structure rather than merely improving formatting.
3. How does the method behave for much longer CoTs or higher compression scales? The paper mainly uses 8 latent tokens and shows a non-monotonic trend when token count increases. Can the authors provide evidence on substantially longer CoTs, or at least a clearer analysis of why performance degrades at larger latent-token counts? A convincing response would increase my confidence in the broader applicability of the method.

**Limitations:**

Yes.

**Strengths And Weaknesses:**

Strength:

**Soundness**. The method is clearly decomposed into visual text tokenization, latent-token-based LLM training, and latent-space inference. The motivation for L-ImgCoT is consistent with the observed limitation that visual compression may blur domain-specific details. The empirical section is reasonably comprehensive for a conference submission: the authors evaluate across four benchmarks spanning math, commonsense, and logical reasoning, and across three different LLM backbones. The main comparison table is informative and shows that ImgCoT is usually stronger than prior latent reasoning baselines at similar token budgets, while L-ImgCoT often surpasses Full-CoT with fewer reasoning tokens. The paper also includes several supportive analyses beyond the main table, including image-vs-text compression, w/o-layout ablation, generalization, and the effect of adding visual latent tokens before retained key steps. Overall, the core empirical claims are supported by a nontrivial amount of evidence.

**Presentation**. The submission is clearly written and well structured. The central intuition is easy to follow, and Figure 1 is effective in communicating the contrast among text-based compression, image-based compression, and the hybrid refinement used in L-ImgCoT. The method section is organized cleanly, and the experimental section is framed around four explicit research questions, which improves readability.

**Originality**. The idea of using rendered CoT images rather than text as the reconstruction target is original and interesting. The paper is not merely proposing another latent reasoning compressor; it reframes the problem through the lens of inductive bias and argues that spatial structure is a better compression target than linguistic surface form. This is a meaningful conceptual shift. The hybrid extension L-ImgCoT is also sensible: it acknowledges the limitation of purely visual compression and offers a practical mechanism for retaining difficult steps.

**Significance**. This work contributes a new direction that may be useful not only for reasoning efficiency but also for understanding what kinds of latent structure are most helpful for reasoning. The generalization experiment is especially interesting because it suggests that the benefit of spatial inductive bias may extend beyond in-domain accuracy and may relate to transfer of abstract reasoning patterns.

Weakness:
1. The γ-based filtering strategy in L-ImgCoT is under-analyzed.
The retained textual steps are chosen using a threshold γ computed from average token log-likelihood on a general pretraining corpus, and steps with lower confidence are kept. This is a reasonable heuristic, but the paper does not sufficiently analyze the sensitivity of the method to γ, nor does it show how different γ choices affect retained-step ratios, final token budget, or task performance. Since γ is central to the proposed hybrid design, an ablation over this threshold would materially strengthen the paper.
2. The paper does not sufficiently disentangle what aspect of “layout” is actually important. The w/o-layout ablation is useful, but it only shows that explicit symbolic cues such as arrows help compared with plain rendering. It remains unclear whether the gain comes primarily from arrow symbols encoding dependency, from line/segment separation itself, or from some broader formatting regularization introduced by the rendering pipeline. Because the paper’s central claim is about spatial inductive bias and structural dependency preservation, a more controlled analysis of these individual layout factors would make the causal interpretation much stronger.
3. Scalability with respect to latent token count remains insufficiently validated. The main setting fixes the number of visual latent tokens to 8, and the paper shows comparisons as the token count varies. However, the broader practical question is whether this style of compression remains effective when the original CoT is much longer, e.g., thousands of tokens rather than a few hundred. The appendix notes a non-monotonic trend when the latent token count becomes large and hypothesizes underfitting, but this analysis is still preliminary. As a result, the paper does not yet fully establish whether the proposed approach will remain strong in very long reasoning settings.
4. There are a few presentation issues and minor inconsistencies. I noticed at least one clear typo around “Table Table 1,” and the reference around line 435 appears confusing and should likely refer to Table 3.

---

> ### Author Rebuttal · Authors · 2026-03-29
>
> We sincerely thank the reviewer for the constructive comments. Below we address all raised concerns with new analyses, additional experiments, and clarified explanations.
> ## 1. Analysis of the γ-based Filtering Threshold
> γ is central to balancing accuracy and token budget. We perform a dedicated ablation in Table 1 and 2.
> ### (a) Sensitivity to γ.
> We vary γ around its default value—the average token log-likelihood over general pretraining corpus—and measure (i) retained-step ratio, (ii) token count, and (iii) accuracy.
>
> ● Higher γ (looser filtering): More steps are retained, increasing token cost. However, performance remains essentially unchanged,  indicating that including additional steps beyond a certain point does not further improve reasoning quality.
>
> ● Lower γ (stricter filtering): Fewer steps are preserved, reducing cost. Once γ falls below the default value, accuracy drops, showing that informative steps are being removed.
>
> **Conclusion: The default γ corresponds to a natural transition between sufficient and insufficient step retention. This supports deriving γ from corpus statistics rather than tuning it per task.** Of course, reducing r to improve efficiency is feasible, but it will make accuracy drop.
>
> #### Table 1 Effects of different values ​​of  γ. Dataset=MATH and LLM=Qwen2.5-0.5B-Instruction.
>
> | γ | ACC | Retained-step (%) | \#Token |
> |---|---|---|---|
> | -2.5886 | 9.8 | 47.6 | 77.6 |
> | -2.0886 | 9.7 | 59.4 | 89.4 |
> | **-1.5886 (default)** | **10.1** | **75.8** | **102.8** |
> | -1.0886 | 10.1 | 91.7 | 139.2 |
> | -0.5866 | 10.0 | 98.05 | 157.3 |
>
> ---
>
> #### Table 2 Effects of different values ​​of  γ. Dataset= MATH and and LLM=Llama3.2-3B-Instruction.
> | γ | ACC | Retained-step (%) | \#Token |
> |---|---|---|---|
> | -2.76450461 | 23.8 | 48.9 | 117.9 |
> | -2.26450461 | 24.1 | 59.4 | 134.4 |
> | **-1.76450461 (default)** | **24.3** | **73.9** | **143.8** |
> | -1.26450461 | 24.3 | 89.9 | 186.2 |
> | -0.76450461 | 24.4 | 98.9 | 216.9 |
>
> ### (b) Stability across corpora.
> To address concerns about corpus dependence, we compute γ on several general-domain pretraining corpora in Table 3. γ remain nearly identical for the same model, demonstrating that **γ is effectively corpus-agnostic**.
>
> ####  Table 3 Default γ across different corpora
>
> | Model | MathPile | dolma | Red\-Pajama |
> |---|---|---|---|
> | Qwen2.5-0.5B-Instruction | -1.58864506 | -1.64774219 | -1.72894371 |
> | Llama3.2-3B-Instruction | -1.76450461 | -1.77974618 | -1.79100659 |
> ---
> ## 2. Contribution of Layout Components
> We conduct controlled ablations in Table 4 to disentangle layout factors using:
>
> ●w/o segmentation: All steps rendered continuously.
>
> ●w/ Random arrows: Arrows preserved but their directions randomized.
>
> ●w/o boxes: Steps rendered without bounding boxes.
>
> **Conclusion : Segmentation is the dominant factor, as it exposes the logical structure of the CoT. Arrows further complement segmentation by indicating dependency direction. Bounding boxes offer only marginal gains, mainly improving visual clarity.**
>
> #### Table 4 Accuracy under Different Rendering Settings
>
> | Dataset | Model | ImgCoT | w/o segmentation | w/o boxes | w/ random arrows |
> |---|---|---|---|---|---|
> | MATH | Qwen2.5-0.5B-Instruction | 9.8 | 9.0 | 9.8 | 9.2 |
> | GPQA | Qwen2.5-0.5B-Instruction | 34.5 | 30.9 | 34.5 | 32.7 |
> | MATH | Llama3.2-3B-Instruction | 24.1 | 23.0 | 23.9 | 23.4 |
> ---
> ## 3. Long CoTs and Latent-Token Scaling Behavior
>
> ### (a) Behavior on longer CoTs.
> We split the MATH test set by length of manual annotated CoT and observe that ImgCoT exhibits only mild performance degradation on tasks that need longer CoT in Table 5. **This shows that the compression mechanism remains effective well beyond the few-hundred-token range in the main experiments.**
>
> ####  Table 5 Acc with ImgCoT on different length of CoT. LLM = Qwen-2.5-0.5B-Instruction
> | Model | <=1000 | >1000 |
> |---|---|---|
> |Qwen-2.5-0.5B-Instruction| 9.8 | 10.3 |
> |LLama3.2-3B-Instruction| 24.1 | 23.1 |
> ### (b) Why performance degrades with more latent tokens.
> Let n be the number of latent tokens. With a fixed codebook size k, the space of latent-token sequences grows combinatorially as C_k^n. SFT only observes at most num_c sequences—bounded by the dataset size. As n increases, num_c / C_k^n quickly shrinks, so the model covers only a vanishing fraction of the latent space, causing underfitting. This explains the non-monotonic trend in Appendix Fig. 8. **Datasets with larger num_c show the performance drop at larger n, further supporting underfitting as the key factor.** A summary table is provided below.
> #### Table 6 (num_c) vs. (n at performance drops)
> | Dataset | \# Training Samples | n at Performance Drop |
> |---|---|---|
> | GPQA | 435 | 4 |
> | GSM | 7500 | 8 |
> | MATH | 7500 | 8 |
> | ProsQA | 17886 | 16 |
> ---
> ## 4. Presentation Issues
> We apologize for these errors and will perform a thorough revision to ensure clarity.
>
> ---
> **We hope this response can ultimately improve your overall assessment.**

---

> > ### Author Rebuttal · Reviewer_fFbf · 2026-04-02
> >
> > The authors have addressed my concerns. I will maintain my positive score.

---

> > > ### Author Response · Authors · 2026-04-02
> > >
> > > We warmly thank the reviewer for the encouraging comments and for noting that the issues have been satisfactorily addressed. The constructive insights provided by the reviewer have played a significant role in improving the manuscript.

---

### Decision · Program_Chairs · 2026-04-30

**Decision:**

Accept (regular)

**Comment:**

This paper proposes ImgCoT, which compresses text CoT into latent CoT through visual space. The approach involves first learning an image tokenizer on rendered CoT, and then finetuning the LLM to work with this latent space. The approach is found to outperform baselines that work purely in text space, as well as other recent "latent CoT" baselines, while being more efficient in terms of the number of "tokens".

The reviewers found the method novel, and supported by empirical gains. There were some concerns about the fact that some of the benchmarks were at saturation (e.g., ProsQA), the small scale of the experiments, as well as questions about limitations of the method (e.g., sensitivity of the approach to the rendering strategy). These were largely addressed in the rebuttal.

I share reviewer jC9i's question/concern about there needing to be a more thorough study of *why* this method works. In particular, both ImgCoT and the text-only version are trained on the same information, and I am not fully convinced that the "inductive bias" of image rendering is what enables the gains. Despite some of these questions that remain, I think this is an interesting nugget of work that could spark more investigations into image-based CoT on rendered data, and hence am recommending acceptance.